# Competitive Co-Evolutionary Learning on Matrix Games with Bandit Feedback

## Abstract

Learning in games is a fundamental problem in machine learning and artificial intelligence, with many successful applications (Silver et al., 2016; Schrittwieser et al., 2020). We consider the problem of learning in matrix games, where two players engage in a two-player zero-sum game with an unknown payoff matrix and bandit feedback. In this setting, players can observe their actions and the corresponding (noisy) payoffs at each round. This problem has been studied in the literature, and several algorithms have been proposed to address it (O'Donoghue et al., 2021; Maiti et al., 2023; Cai et al., 2023). In particular, O'Donoghue et al. (2021) demonstrated that deterministic optimism (e.g., the UCB algorithm for matrix games) plays a central role in achieving sublinear regret and outperforms other algorithms. However, despite numerous applications, the theoretical understanding of learning in matrix games remains underexplored. Specifically, it remains an open question whether randomised optimism can also exhibit sublinear regret.

In this paper, we propose a novel algorithm called Competitive Co-evolutionary Bandit Learning (COEBL) for unknown two-player zero-sum matrix games. By integrating evolutionary algorithms (EAs) into the bandit framework, COEBL introduces randomised optimism through the variation operator of EAs. We prove that COEBL also enjoys sublinear regret, matching the regret performance of algorithms based on deterministic optimism (O'Donoghue et al., 2021). To the best of our knowledge, this is the first work that provides a regret analysis of an evolutionary bandit learning algorithm in matrix games. Empirically, we compare COEBL with classical bandit algorithms, including EXP3 (Auer et al., 2002), the variant of EXP3-IX (Cai et al., 2023), and UCB algorithms analysed in O'Donoghue et al. (2021) across several matrix game benchmarks. Our results show that COEBL not only enjoys sublinear regret, but also outperforms existing methods in various scenarios. These findings reveal the promising potential of evolutionary bandit learning in game-theoretic settings, in particular, the effectiveness of randomised optimism via evolutionary algorithms.

## 1 Introduction

### 1.1 Two-Player Zero-Sum Games

Triggered by Von Neumann's seminal work (Von Neumann, 1928; Von Neumann et al., 1953), the maximin optimisation problem (i.e., $\max_{x \in \mathcal{X}} \min_{y \in \mathcal{Y}} g(x, y)$) has become a major research topic in machine learning and optimisation. In particular, two-player zero-sum games, represented by a payoff matrix $A \in \mathbb{R}^{m \times m}$, are a popular class of problems explored in much of the current machine learning and AI literature (Littman, 1994; Auger et al., 2015; O'Donoghue et al., 2021; Cai et al., 2023). The row player selects $i \in [m]$, the column player selects $j \in [m]$ and these choices, leading to a payoff $A_{ij}$ (i.e. the row player receives the payoff $A_{ij}$ and the column player receives the payoff $-A_{ij}$). Generally, we are interested in finding the optimal mixed strategy, which is the probability distribution over all actions for each player. Thus, we can formulate our problem as follows: to find $x^*, y^* \in \Delta_m$, where $\Delta_m$ denotes the probability simplex of dimension $m - 1$, satisfying

$$V_A^* := \max_{x \in \Delta_m} \min_{y \in \Delta_m} y^T A x. \tag{1}$$

By minimax theorem (Von Neumann, 1928), $V_A^* = \min_{y \in \Delta_m} \max_{x \in \Delta_m} y^T A x$. $(x^*, y^*)$ solving for Eq.(1) is also called Nash equilibrium. $V_A^*$ is the shared optimal quantity at the Nash equilibrium. In this paper, we call it the Nash equilibrium payoff.

Nash's Theorem, or the Minimax Theorem, guarantees the existence of $(x^*, y^*)$ for Eq.(1) (Von Neumann, 1928; Nash, 1950). If the payoff matrix is given or known, then Eq.(1) can be reformulated as a linear programming problem, and it can be solved in polynomial runtime using algorithms including the ellipsoid method or interior point method (Bubeck et al., 2015; Maiti et al., 2023). Now, if the payoff matrix is unknown, let the row and column player play an iterative two-player zero-sum game. At each iteration, based on their action, we can query one of the entries in the payoff matrix, and then both players can adjust their strategies based on the observed payoff (or reward). We repeat these iterations until the stopping criteria are met. This kind of two-player zero-sum game is called repeated matrix games (or matrix games, for short). Our interest lies in algorithms that can outperform others in matrix games. One of the common metrics measuring the performance of algorithms in matrix games is regret, which will be defined in later sections. We are also interested in whether they can find or approximate the Nash equilibrium $(x^*, y^*)$, as measured by metrics such as KL-divergence or total variation distance.

## 1.2 EVOLUTIONARY REINFORCEMENT LEARNING AND COEVOLUTION

Evolutionary Algorithms (EAs) are randomised heuristics that mimic natural selection to solve optimisation problems (Popovici et al., 2012; Eiben & Smith, 2015). EAs aim to find global optima with minimal knowledge about fitness functions, making them well-suited for black-box or oracle settings compared to gradient-based methods. They are powerful tools for discovering effective reinforcement learning policies. EAs are particularly useful because they can identify good representations, manage continuous action spaces, and handle partial observability. Due to these strengths, evolutionary reinforcement learning (ERL) techniques have shown strong empirical success and we refer readers to (Whiteson, 2012; Bai et al., 2023; Li et al., 2024a) for detailed reviews of ERL.

Coevolution, a concept from evolutionary biology, describes the interactions between individuals evolving together. It occurs when an individual's fitness depends on others also evolving (Popovici et al., 2012). Coevolution can be either cooperative, such as the relationship between humans and gut bacteria, or competitive, like predator-prey dynamics. These coevolutionary dynamics have been studied and applied in ERL, demonstrating empirical effectiveness in many applications (Whiteson, 2012; Xue et al., 2024; Li et al., 2024a). For example, co-evolutionary algorithms (CoEAs), a subset of EAs, have been applied in many black-box optimisation problems under various game-theoretic optimisation scenarios (Xue et al., 2024; Gomes et al., 2014; Hemberg et al., 2021; Flores et al., 2022; Fajardo et al., 2023; Hevia Fajardo et al., 2024; Benford & Lehre, 2024a).

Evolutionary reinforcement learning has achieved great success in many applications, including game playing, robotics, and optimisation (Moriarty et al., 1999; Khadka & Tumer, 2018; Pourchot & Sigaud, 2019; HAO et al., 2023; Li et al., 2024b;c), but there is barely any theoretical analysis of these powerful methods. In particular, the theoretical understanding of coevolutionary learning remains blanked, especially in the context of matrix games. As a starting point, in this paper, we combine evolutionary heuristics with bandit learning and explore this combination in matrix games from both theoretical and empirical perspectives.

## 1.3 CONTRIBUTIONS

This paper introduces evolutionary algorithms for learning in matrix games with bandit feedback. To the best of our knowledge, this is the first paper to rigorously analyse the regret of evolutionary reinforcement learning (i.e., COEBL) for matrix games with bandit feedback. Specifically, we demonstrate that randomised optimism via evolution can also exhibit sublinear regret in matrix games. Our empirical results show that COEBL outperforms the existing bandit baselines for matrix games, including EXP3, UCB, and the EXP3-IX variant. These findings highlight the great potential of evolutionary algorithms for bandit learning in matrix games and reveal the importance of randomness in game playing. It serves as the first step towards rigorously theoretical understanding of evolutionary reinforcement learning.

### 1.4 RELATED WORKS

#### 1.4.1 REGRET ANALYSIS OF BANDIT LEARNING IN MATRIX GAMES

Theoretical analysis of bandit learning algorithms in matrix games has been extensively studied. Recent works, such as (Auger et al., 2015; O'Donoghue et al., 2021; Cai et al., 2023), have examined classical bandit algorithms in matrix games, where only rewards or payoffs are observed. In particular, O'Donoghue et al. (2021) conducted a detailed regret analysis on the UCB algorithm, Thompson Sampling, and K-Learning [1]. They showed sublinear regret bounds for these existing bandit baselines in matrix games. Neu (2015) proved a sublinear regret bound for EXP3-IX, and later, Cai et al. (2023) proposed a new variant of EXP3-IX for matrix games. Additionally, Auger et al. (2015) convergence analyses of bandit algorithms on sparse binary zero-sum games, while Cai et al. (2023) extended the convergence analysis to uncoupled bandit learning in two-player zero-sum Markov games. However, the theoretical analysis of evolutionary reinforcement learning remains largely unexplored. Our work aims to address this gap, marking the first step toward understanding evolutionary bandit learning in matrix games, an exciting and under-explored area.

#### 1.4.2 RUNTIME ANALYSIS OF COEVOLUTIONARY ALGORITHMS ON GAMES

Recent studies have conducted runtime analyses of coevolutionary algorithms in two-player zero-sum games (Jansen & Wiegand, 2004; Lehre, 2022; Hevia Fajardo & Lehre, 2023; Fajardo et al., 2023; Lehre & Lin, 2024; Benford & Lehre, 2024a;b). In this context, runtime refers to the number of function evaluations required by the algorithms to find the Nash equilibrium. For a more detailed introduction to these works, we refer readers to the recent paper by Benford & Lehre (2024b). While we do not analyse the runtime of COEBL in this paper, it would be interesting to explore how the runtime of COEBL could be studied in the context of matrix games with bandit feedback in future work. The idea of competitive coevolution in game-theoretic settings is derived from the works mentioned in this section, which we apply to the bandit learning in matrix games.

### 2 PRELIMINARIES

#### 2.1 NOTATIONS

Given $n \in \mathbb{N}$, we write $[n] := \{1, 2, \cdots, n\}$. $\mathbb{F}_p$ denotes the finite field of $p$ (prime number) elements. For example, $\mathbb{F}_3$ denotes the finite field of three elements, $\{-1, 0, 1\}$. We denote the row player by the $x$-player and the column player by the $y$-player. $f(n) \in O(h(n))$ if there exists some constant $c > 0$ such that $f(n) \leq ch(n)$. $f(n) \in \tilde{O}(h(n))$ if there exists some constant $k > 0$ such that $f \in O(h(n) \log^k (h(n)))$. We define the $(m - 1)$-dimensional probability simplex as $\Delta_m := \{z \in \mathbb{R}^m \mid \sum_{i=1}^m z_i = 1, z_i \geq 0\}$. In each round $t \in \mathbb{N}$, the row player chooses $i_t \in [m]$, and the column player chooses $j_t \in [m]$; and then $r_t$ is the reward obtained by the row player. We define the corresponding filtration $\mathcal{F}_t$ prior to round $t$ by $\mathcal{F}_t := (i_1, j_1, r_1, \ldots, i_{t-1}, j_{t-1}, r_{t-1})$. We denoted $\mathrm{E}_t(\cdot) := \mathrm{E}(\cdot \mid \mathcal{F}_t)$. For any real number $x$, we define $1 \vee x := \max(1, x)$. Given $x \in \{0, 1\}^n$, $|x|_1 := \sum_{i=1}^n x_i$.

**Definition 1.** A random variable $X \in \mathbb{R}$ is $\sigma^2$-sub-Gaussian with variance proxy $\sigma^2$ if $\mathrm{E}(X) = 0$ and its moment generating function satisfies $\mathrm{E}(\exp(sX)) \leq \exp\left(\frac{\sigma^2 s^2}{2}\right)$, for all $s \in \mathbb{R}$.

#### 2.2 P-ARY TWO-PLAYER ZERO-SUM GAMES AND NASH REGRET

A two-player game is defined by the strategy spaces $\mathcal{X}$ and $\mathcal{Y}$, along with payoff functions $g_i : \mathcal{X} \times \mathcal{Y} \to \mathbb{R}$, where $i \in [2]$. Here, $g_i(x, y)$ represents the payoff to player $i$ when player 1 plays strategy $x$ and player 2 plays strategy $y$.

**Definition 2.** Given a two-player game with strategy spaces $\mathcal{X}$ and $\mathcal{Y}$ and prime number $p \in \mathbb{N}$, the payoff functions $g_1, g_2 : \mathcal{X} \times \mathcal{Y} \to \mathbb{R}$ are defined for player 1 and player 2, respectively. The game is zero-sum if player 1's gain is exactly equal to player 2's loss (and vice versa), meaning

---

[1]UCB has been shown to outperform the other two algorithms in (O'Donoghue et al., 2021); hence, we consider UCB as our primary baseline for matrix games in this paper.

$g_1(x, y) + g_2(x, y) = 0$ for all $x \in \mathcal{X}$ and $y \in \mathcal{Y}$. If $g_1(x, y), g_2(x, y) \in \mathbb{F}_p$ for all $x, y \in \mathcal{X}$, the game is called $p$-ary (i.e. binary if $p = 2$, ternary if $p = 3$ and quinary if $p = 5$). Given a payoff function $g : \mathcal{X} \times \mathcal{Y} \to \mathbb{F}_p$, we refer to the game with payoff functions $g_1(x, y) = g(x, y)$ and $g_2(x, y) = -g(x, y)$ as the *$p$-ary zero-sum game* defined by $g$.

Many classical games where the outcomes are win/lose/draw, such as Rock-Paper-Scissors, Tic-Tac-Toe and Go, can be modelled as ternary zero-sum games by identifying $g(x, y) = 1$ with a win for player 1, $g(x, y) = -1$ with a win for player 2, and $g(x, y) = 0$ with a draw. In this paper, we mainly focus on ternary two-player zero-sum games.

In matrix games, we consider the Nash regret as our performance measure, defined as the cumulative difference between the Nash equilibrium payoff in Eq.(1) and the rewards obtained by the players.

**Definition 3** (Nash Regret (O'Donoghue et al., 2021)). Consider any matrix game with payoff matrix $A \in \mathbb{R}^{m \times m}$ and the reward for the row player choosing action $i_t \in [m]$ and the column player choosing action $j_t \in [m]$ is given by $r_t = A_{i_t j_t} + \eta_t$, where $\eta_t$ is zero-mean noise, independent and identically distributed from a known distribution at iteration $t \in \mathbb{N}$. Given an algorithm ALG that maps the filtration $\mathcal{F}_t$ to a distribution over actions $x \in \Delta_m$, we define the Nash regret with respect to the Nash equilibrium payoff $V_A^* \in \mathbb{R}$ by

$$\mathcal{R}(A, \text{ALG}, T) := \mathrm{E}_{\eta, \text{ALG}} \left( \sum_{t=1}^{T} V_A^* - r_t \right).$$

Given any class of games $A \in \mathcal{A}$, we define
$$\text{WORSTCASEREGRET}(\mathcal{A}, \text{ALG}, T) := \max_{A \in \mathcal{A}} \mathcal{R}(A, \text{ALG}, T).$$

Given a fixed unknown payoff matrix $A$, $\mathcal{R}(A, \text{ALG}, T)$ represents the expected cumulative difference between the Nash equilibrium payoff and the rewards obtained by player 1 using ALG over $T$ iterations. WORSTCASEREGRET $(\mathcal{A}, \text{ALG}, T)$ considers the maximum regret of Algorithm ALG over all the possible payoff matrices in the class of games $\mathcal{A}$. In other words, it denotes the expected regrets under the worst-case scenario.

## 3 CO-EVOLUTIONARY BANDIT LEARNING FOR MIXED NASH EQUILIBRIUM

### 3.1 LEARNING IN GAMES AND COEBL

Many studies have analysed how players learn to reach equilibrium when playing against opponents (Fudenberg & Levine, 1998). Briefly speaking, learning in games aims to understand how a player can learn to reach or approximate equilibrium and win the games when playing against either rational or irrational opponents. One of the common measures to evaluate the performance of algorithms in games is regret, which is defined in Definition 3. Other measures include convergence to Nash equilibrium, which can be measured by KL-divergence or total variation distance.

In this paper, we only present the algorithm for the $x$-player as the algorithm for the opponent is symmetric. We defer other algorithms to the supplementary material and only present the proposed algorithm in this section. COEBL stands for Co-Evolutionary Bandit Learning. We use $\bar{A}_{ij}^t$ to denote the empirical mean of the samples from $A_{ij}$ and $n_{ij}^t \in [t] \cup \{0\}$ to denote the number of times the row player chooses action $i$ and the column player chooses $j$ up to round $t$.

The following mutation variant is considered in this paper.

$$\text{Mutate}(\bar{A}_{ij}^t, \frac{1}{1 \vee n_{ij}^t}) = \bar{A}_{ij}^t + \mathcal{N}\left( \sqrt{\frac{c \log(2T^2 m^2)}{1 \vee n_{ij}^t + 1}}, \frac{1}{(1 \vee n_{ij}^t)^2} \right),$$

where $\mathcal{N}(\mu, \sigma^2)$ denotes a Gaussian random variable with mean $\mu$ and variance $\sigma^2$, and $c$ is some constant with respect to $T$ and $m$.

Evolutionary algorithms consist of two main components: variation operators and selection mechanisms. Variation operators can generate new individuals from the current population, and the selection mechanism chooses the best individuals from the population based on the fitness function. In

---

**Algorithm 1** COEBL for matrix games

---

**Require:** Fitness function: $\text{Fitness}(x, B) := \min_{y \in \Delta_m} y^T B x$ where $B \in \mathbb{R}^{m \times m}$ and $x \in \Delta_m$.
 1: **Initialisation:** $x_0, y_0 = (1/m, \ldots, 1/m)$ and $n_{ij}^0 = 0$ for all $i, j \in [m]$
 2: **for** round $t = 1, 2, \ldots, T$ **do**
 3:     **for** all $i, j \in [m]$ **do**
 4:         Compute $\tilde{A}_{ij}^t = \text{Mutate}(\bar{A}_{ij}^t, 1/1 \vee n_{ij}^t)$
 5:     **end for**
 6:     Obtain the mutated policy $x' \in \arg\max_{x \in \Delta_m} \min_{y \in \Delta_m} y^T \tilde{A}^t x$
 7:     **if** $\text{Fitness}(x', \tilde{A}^t) > \text{Fitness}(x_{t-1}, \tilde{A}^t)$ **then**
 8:         Update policy $x_t := x'$
 9:     **else**
10:         Update policy $x_t := x_{t-1}$
11:     **end if**
12:     Update the query number of each entry in the payoff matrix $n_{ij}^t$ for all $i, j \in [m]$
13: **end for**

---

COEBL, the fitness function $\text{Fitness}(x, B) := \min_{y \in \Delta_m} y^T B x$ is used to evaluate the performance of policy $x$ against the best response of the opponent given payoff matrix $B$. At the beginning, COEBL employs a Gaussian mutation operator to generate a new estimated payoff matrix $\tilde{A}^t$ and then the mutated policy $x'$ for the row player. Note that, since the estimated payoff matrix $\tilde{A}$ is always fully accessible to the $x$-player, $x'$ in line 6 can be obtained by solving a linear programming problem (Bubeck et al., 2015; Maiti et al., 2023). Between lines 7 and 10, we use the fitness function to evaluate the performance of policy $x'$ and compare it with the previous policy $x_{t-1}$. If the new policy $x'$ strictly outperforms the previous policy $x_{t-1}$, we update the policy $x_t$ to $x'$; otherwise we keep the policy $x_t$ as $x_{t-1}$.

The main idea of COEBL is to employ the principle of 'optimism in the face of uncertainty' (OFU) to explore the action space and exploit the opponent's best response (Bubeck et al., 2012; Lattimore & Szepesvári, 2020). However, the main difference between COEBL and other bandit algorithms including the original UCB family is that COEBL adopts randomised optimism through the use of evolutionary algorithms. As a result, via the variation operator, COEBL can generate more diverse estimated payoff matrices, leading to more diverse policies. Then, the selection mechanisms guide the evolutionary process towards higher fitness. Unlike (O'Donoghue et al., 2021), which mentioned that deterministic optimism plays a central role in enabling UCB (Algorithm 4) to exhibit sublinear regret and outperform the classic EXP3 and other bandit baselines, we will show that randomised optimism (via evolution) also exhibits sublinear regret. Furthermore, we will demonstrate that randomised optimism in matrix games can be more effective and adaptive in preventing exploitation by the opponent than deterministic optimism, and thus outperforms existing bandit baselines. Specifically, it outperforms the current bandit baseline algorithms for matrix games, including EXP3 (Algorithm 3), UCB (Algorithm 4) and the EXP3-IX variant (Algorithm 5) on the matrix game benchmarks discussed in this paper.

### 3.2 REGRET ANALYSIS OF COEBL

In this section, we conduct the regret analysis of COEBL in matrix games. Before our analysis, we need some technical lemmas. We defer these lemmas to the appendix.

We follow the setting in (O'Donoghue et al., 2021) and consider the case where there is 1-sub-Gaussian noise when querying the payoff matrix. Assume the following, given $t \in \mathbb{N}$:

      (A): The noise process $\eta_t$ is 1-sub-Gaussian and the payoff matrix satisfies $A \in [0, 1]^{m \times m}$.

**Lemma 1.** *Suppose Assumption (A) holds with $T \geq 2m^2 \geq 2$ and $\delta := \left(1/2T^2 m^2\right)^{c/8}$ where $c > 0$ is the mutation rate in COEBL. For each iteration $t \in \mathbb{N}$, given $\tilde{A}^t$ in Algorithm 1, we have:*

$$\Pr\left(A_{ij} - (\tilde{A}_t)_{ij} \leq 0\right) \geq 1 - \delta, \quad \text{for all } i, j \in [m]. \tag{2}$$

**Theorem 2** (Main Result). *Consider any two-player zero-sum matrix game. Under Assumption (A) with $T \geq 2m^2 \geq 2$ and $\delta = \left(1/2T^2m^2\right)^{c/8}$, where $c > 0$ is the mutation rate in COEBL, the worst-case Nash regret of COEBL for $c \geq 8$ is bounded by $2\sqrt{2cTm^2 \log(2T^2m^2)}$, i.e., $\tilde{O}(\sqrt{m^2T})$.*

*Sketch of Proof.* Due to page limit, we defer the full proof to the appendix and provide a simple proof sketch here. First, we bound the regret under the case where all the entries of the estimated payoff matrix are greater than those of the real, unknown payoff matrix (this event is denoted by $E_t^c$ at iteration $t \in \mathbb{N}$). Secondly, we use the law of total probability to consider both cases: when all the entries of the estimated payoff matrix are greater than the real payoff matrix, and the converse (i.e., event $E_t$). We already have the upper bound for the first part; the second part can be trivially bounded by 1 in each iteration. Using Lemma 1, we can obtain the upper bound of probability of event $E_t$. Combining these bounds provides us with the upper bound for the regret of COEBL. □

Theorem 2 shows that, under the worst-case scenario (assuming the best response of the opponent across all the possible matrix game instances under Assumption (A)), COEBL can also exhibit sublinear regret. More precisely, the regret of COEBL is bounded by $\tilde{O}(\sqrt{m^2T})$, which is the same as the regret bound of UCB. This implies that deterministic optimism in the face of uncertainty is not the crucial factor for achieving sublinear regret, as discussed in (O'Donoghue et al., 2021). The current results considers $c \geq 8$ in the analysis due to current technical limitations. We conjecture that the regret bound can be improved by considering smaller $c$ values, and thus, in practical use, we suggest one may need hyperparameter tuning in various problems. Additionally, as we will show later, randomised optimism via evolution can be more robust than deterministic optimism in game playing, and therefore COEBL outperforms the other algorithms in the following benchmarks.

## 4 EMPIRICAL RESULTS

In this section, we present empirical results comparing the discussed algorithms. We are interested in empirical regret in specific game instances, measured by cumulative (absolute) regret, i.e.,

$$\sum_{t=1}^{T} |V_A^* - r_t| \quad \text{and} \quad \sum_{t=1}^{T} V_A^* - r_t \tag{3}$$

where $r_t$ is the obtained reward at round $t$. We focus on two scenarios, including self-play and ALG 1-vs-ALG 2. In the self-play scenario, both row and column players use the same algorithm with the same information. We use the absolute regret (the first metric) to measure the performance of the algorithms in this case. The ALG 1-vs-ALG 2 is a generalisation of the self-play scenario. We use the second metric in Eq. 3 to measure the performance of the algorithms. The ALG 1-vs-ALG 2 means the row player uses ALG 1, and the column player uses ALG 2 with the same information. As in the setting of (O'Donoghue et al., 2021), the plots below show the regret (not absolute regret) from the maximiser's (ALG 1) perspective. A positive regret value means that the minimiser (ALG 2) is, on average winning and vice versa. This allows us to compare our algorithms directly.

Moreover, to measure how far the players are from the Nash equilibrium, we use the KL-divergence between the policies of both players and the Nash equilibrium or the total variation distance (for the case where the KL-divergence is not well-defined), i.e.,

$$\text{KL}(x_t, x^*) + \text{KL}(y_t, y^*) := \sum_i x_t(i) \ln\left(\frac{x_t(i)}{x^*(i)}\right) + \sum_j y_t(j) \ln\left(\frac{y_t(j)}{y^*(j)}\right)$$

$$\text{TV}(x_t, x^*) + \text{TV}(y_t, y^*) := \frac{1}{2}\sum_i |x_t(i) - x^*(i)| + \frac{1}{2}\sum_j |y_t(j) - y^*(j)|$$

where $(x^*, y^*)$ is the Nash equilibrium of $A$.

**Parameter Settings**: Given $K$ is the number of actions for each player and $T$ is the time horizon, for EXP3, we use the exploration rate $\gamma_t = \min\{\sqrt{K \log K / T}, 1\}$ and learning rate $\eta_t = \sqrt{2 \log K / TK}$ as suggested in O'Donoghue et al. (2021). For the variant of EXP3-IX, we use the same settings $\eta_t = t^{-k_\eta}, \beta_t = t^{-k_\beta}, \varepsilon_t = t^{-k_\varepsilon}$ where $k_\eta = \frac{5}{8}, k_\beta = \frac{3}{8}, k_\varepsilon = \frac{1}{8}$ as suggested in

Cai et al. (2023). For COEBL, we set the mutation rate $c = 2$ for the RPS game and $c = 8$ for the rest of the games. There is no hyperparameter needed for UCB. For the observed reward, we consider standard Gaussian noise with zero mean and unit variance, i.e. $r_t = A_{i_t, j_t} + \eta_t$ where $\eta_t \sim \mathcal{N}(0, 1)$. We compute the empirical mean of the regrets and the KL-divergence (or total variation distance), and present the 95% confidence intervals in the plots. We run 50 independent simulations for each configuration (over 50 seeds).

## 4.1 ROCK-PAPER-SCISSORS GAME

We consider the classic matrix game benchmark: Rock-Paper-Scissors games (Littman, 1994; O'Donoghue et al., 2021), and its payoff matrix is defined as follows.

|   | R | P | S |
|---|---|---|---|
| R | 0 | 1 | -1 |
| P | -1 | 0 | 1 |
| S | 1 | -1 | 0 |

Table 1: The payoff matrix of RPS game. R denotes rock, P denotes paper, and S denotes scissors.

It is well known that $x^*, y^* = (1/3, 1/3, 1/3)$ is the unique mixed Nash equilibrium of the RPS game for both players. We conduct experiments using Algorithms 3 to 5 and compare them with our proposed Algorithm 1 (i.e. COEBL) on the classic matrix game benchmark: the RPS game.

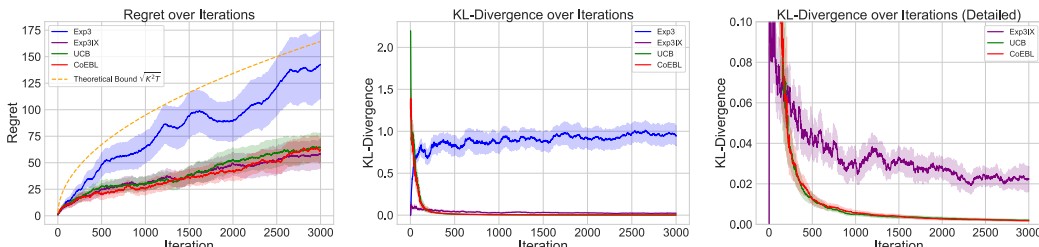

Figure 1: Regret and KL-divergence for Self-Plays on RPS games

In Figure 1, we present the self-play results of each algorithm. We can observe that COEBL also exhibits sublinear regret in the RPS game, similar to other bandit baselines, and matches our theoretical bound. In terms of the KL-divergence, EXP3, as reported in (O'Donoghue et al., 2021; Cai et al., 2023), diverges from the Nash equilibrium. By zooming in on the KL-divergence plot, we can observe that COEBL and UCB converges to the Nash equilibrium faster than the other algorithms; especially, EXP3-IX has a much slower convergence rate.

Next, we compare the performance of the algorithms by examining their regret bounds and KL-divergence from the Nash equilibrium when algorithms compete with each other using the same information. In Figure 2, we can clearly observe that COEBL outperforms the EXP3 family, including EXP3 and EXP3-IX, in terms of regret. On average, COEBL has a smaller advantage over UCB in terms of regret, since the empirical mean of regret is above 5 but below 10 after iteration 2000.

The RPS game with a small number of actions is relative simple for these algorithms to play. Moreover, although COEBL completely outperforms the EXP3 family, it does not have an overwhelming advantage over the UCB. How do these algorithms behave on more complex games with exponentially many actions? Can COEBL still take over the game? Next, we answer these questions by considering DIAGONAL and BIGGERNUMBER games.

## 4.2 DIAGONAL GAME

DIAGONAL is a pseudo-Boolean maximin-benchmark on which Lehre & Lin (2024) conducted runtime analysis of coevolutionary algorithms. Both players have an exponential number (i.e. $2^n$)

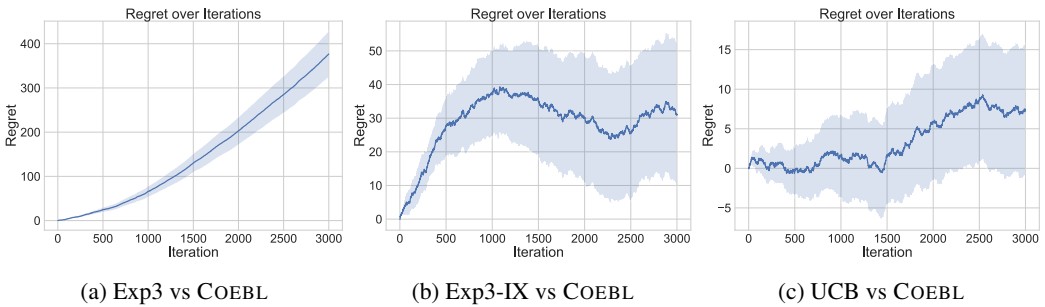

(a) Exp3 vs COEBL      (b) Exp3-IX vs COEBL      (c) UCB vs COEBL

Figure 2: Regret for ALG 1-vs-ALG 2 on RPS games

of pure strategies. To distinguish between pure strategies that consist of the same number of $1$, we modify the original DIAGONAL by introducing a 'draw' outcome. For $\mathcal{U} = \{0,1\}^n$ and $\mathcal{V} = \{0,1\}^n$, the payoff function DIAGONAL $: \mathcal{U} \times \mathcal{V} \to \{0,1\}$ is defined by

$$\text{DIAGONAL}(u,v) := \begin{cases} 1 & |v|_1 < |u|_1 \\ 0 & |v|_1 = |u|_1 \\ -1 & \text{otherwise} \end{cases}.$$

As shown by Lehre & Lin (2024), this game (we provide a simple example in the appendix) exhibits a unique pure Nash equilibrium where both players choose $1^n$. This corresponds to the mixed Nash equilibrium where $x^* = (0, \cdots, 1)$ and $y^* = (0, \cdots, 1)$. We conduct experiments using Algorithms 3 to 5 and compare them with our proposed Algorithm 1 (i.e. COEBL) on another matrix game benchmark: the DIAGONAL game. We set the mutation constant $c = 8$ for COEBL and consider $n = 2, 3, 4, 5, 6, 7$ in the experiments.

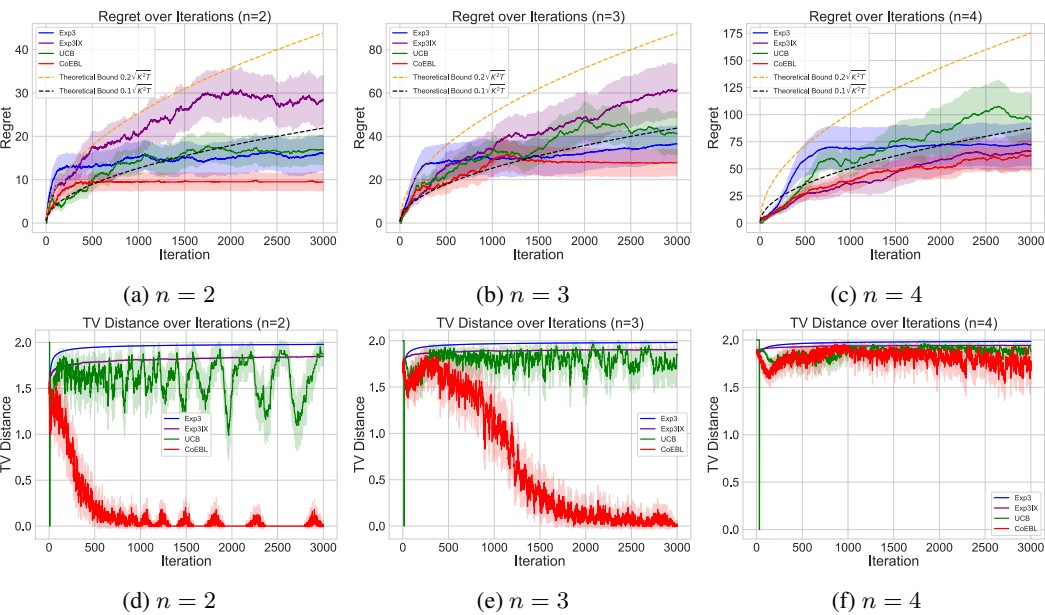

(a) $n = 2$      (b) $n = 3$      (c) $n = 4$

(d) $n = 2$      (e) $n = 3$      (f) $n = 4$

Figure 3: Regret and TV Distance for Self-Plays on DIAGONAL

In Figures 3 and 7, we present the self-play results of each algorithm on DIAGONAL game for various values of $n$. Our results show that COEBL consistently exhibits sublinear regret in the DIAGONAL game, aligning with our theoretical bounds and similar to other bandit algorithms. As $n$ increases, the regret of the baseline algorithms grows as expected. COEBL remains more adaptive and robust in more challenging games, maintaining sublinear regret beneath the theoretical bound ($0.1\sqrt{K^2T}$), as indicated by the black dotted line. We also observe that the regrets of all algorithms increases as

$n$ grows, which is expected due to the exponential increase in the number of pure strategies and the corresponding complexity of the game. In terms of convergence measured by TV-distance, COEBL converges to the Nash equilibrium for $n = 2, 3$, while the baseline algorithms do not converge. However, for $n \geq 4$, as the number of strategies grows exponentially, COEBL also struggles to converge to the Nash equilibrium. In Figures 4 and 8, we present the regrets for ALG 1-vs-ALG 2 on DIAGONAL. The empirical regrets across all algorithms exceed 16.2, with a maximum of 389.8 for $n = 6$, indicating that the minimiser is dominant. In other words, COEBL outperforms the other bandit algorithms across all values of $n$, from 2 to 7.

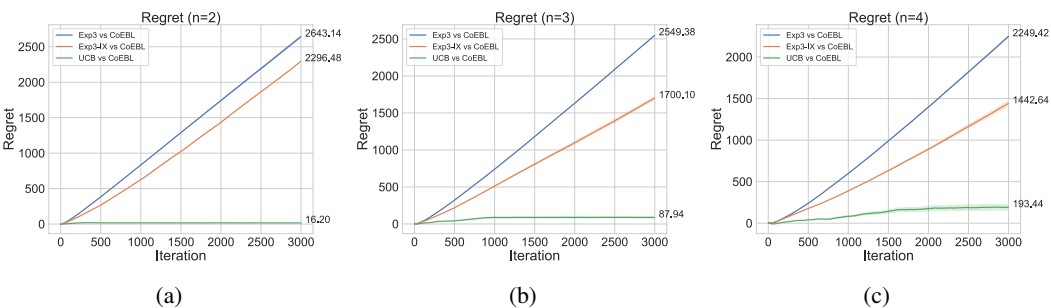

Figure 4: Regret for ALG 1-vs-ALG 2 on DIAGONAL.

### 4.3 BIGGERNUMBER GAME

BIGGERNUMBER is another challenging two-player zero-sum game proposed and analysed by Zhang & Sandholm (2024). In this game, each player aims to select a number that is larger than their opponent's. The players' action space is $\mathcal{X} = \{0, 1\}^n$, representing binary bitstrings of length $n$ corresponding to natural numbers in the range $[0, 2^n - 1]$. A formal definition and the complete results can be found in the appendix. We present part of the results here.

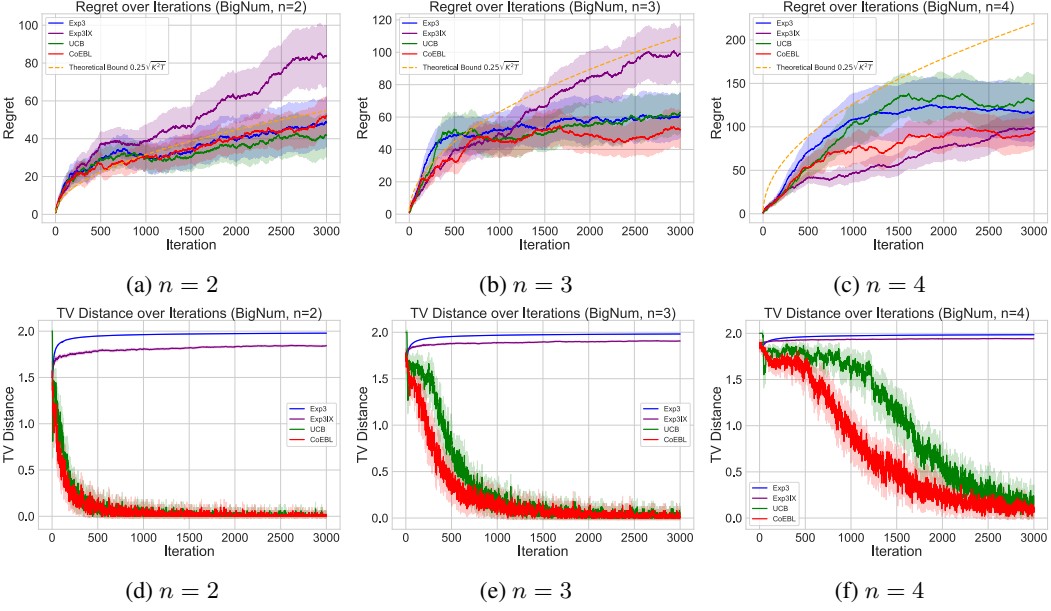

Figure 5: Regret and TV Distance for Self-Plays on BIGGERNUMBER

In summary, we conducted extensive experiments on three matrix games: the RPS game, the DIAGONAL game, and the BIGGERNUMBER game. In terms of regret performance, COEBL in self-play

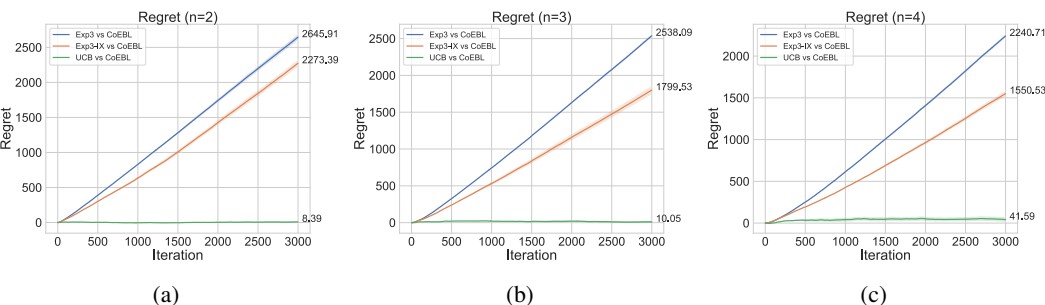

Figure 6: Regret for ALG 1-vs-ALG 2 on BIGGERNUMBER.

aligns with our theoretical bounds. Moreover, COEBL consistently outperforms other bandit baselines when competing across various matrix game benchmarks, as shown in Figures 4 and 6. COEBL matches the performance of UCB and converges more quickly than EXP3-IX in the RPS game. CO-EBL converges to the Nash equilibrium for $n = 2, 3$ and for $n = 2, 3, 4$, respectively, while the other baselines do not converge, as shown in Figures 3 and 5. Therefore, we conclude that COEBL is a promising algorithm for matrix games, demonstrating sublinear regret, outperforming other bandit baselines, and achieving convergence to the Nash equilibrium in several matrix game instances. However, as the number of strategies grows exponentially, COEBL, like other algorithms, fails to converge to the Nash equilibrium. This observation points out the current limitation of existing algorithms in exponentially large matrix games, and it will be an exciting path for future research.

## 5 CONCLUSION AND DISCUSSION

This paper addresses the unsolved problem of learning in unknown two-player zero-sum matrix games with bandit feedback, proposing a novel algorithm, COEBL, which integrates evolutionary algorithms with bandit learning. To the best of our knowledge, this is the first work that combines evolutionary algorithms and bandit learning for matrix games and provides regret analysis of evolutionary bandit learning (EBL) algorithms in this context. This paper demonstrates that randomised or stochastic optimism, particularly through evolutionary algorithms, can also enjoy a sublinear regret in matrix games, offering a more robust and adaptive solution compared to traditional methods.

Theoretically, we prove that COEBL exhibits sublinear regret in matrix games, extending the rigorous understanding of evolutionary approaches in bandit learning. Practically, we show through extensive experiments on various matrix games—including the RPS, DIAGONAL, and BIGGER-NUMBER games that COEBL outperforms existing bandit baselines, offering practitioners a new tool (randomised optimism via evolution) for handling matrix games playing with bandit feedback.

Despite these promising results, our work has some limitations. Theoretically, we only consider two-player zero-sum games, which is consistent with prior studies such as (O'Donoghue et al., 2021; Cai et al., 2023). Extending COEBL to general-sum games with more players or to Markov games represents an exciting and challenging avenue for future research. More technically, we conjecture whether Theorem 2 could also hold for smaller value of $c < 8$ with certain threshold. Additionally, our analysis assumes sub-Gaussian noise; investigating the algorithm's performance under different noise distributions, such as sub-exponential noise, could yield further insights. From an experimental perspective, testing on more diverse problem instances would strengthen the empirical analysis.

Future work could focus on both theoretical and practical extensions of evolutionary bandit learning. From a theoretical perspective, it would be worthwhile to explore how COEBL or other evolutionary bandit learning algorithms can be adapted to more complex game structures, such as multi-player or general-sum games. On the practical side, improving COEBL by incorporating more sophisticated mutation operators, additional crossover operator, non-elitist selection mechanisms, or population-based evolutionary algorithms could enhance its performance in more complex settings.

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
