CONTENTS

# A APPENDIX

## A.1 SUMMARY OF REGRET BOUNDS

We provide a table to summarise the relevant regret bounds for different algorithms in matrix games with bandit feedback.

| Algorithms | HEDGE (Freund & Schapire, 1997) | GP-MW (Sessa et al., 2019) | EXP3 (Auer et al., 2002) | EXP3-IX (Neu, 2015; Cai et al., 2023) | UCB/ K-Learning (O'Donoghue et al., 2021) | COEBL Theorem 2 |
|---|---|---|---|---|---|---|
| Feedback | rewards for all actions | obtained reward + opponents' actions | obtained reward | obtained reward | obtained reward | obtained reward |
| Regret | $\mathcal{O}\left(\sqrt{T \log K}\right)$ | $\mathcal{O}\left(\sqrt{T \log K}\right) + \gamma_T \sqrt{T}$ | $\mathcal{O}\left(\sqrt{TK \log K}\right)$ | $\mathcal{O}\left(\sqrt{TK \log K}\right)$ | $\tilde{\mathcal{O}}\left(\sqrt{K^2 T}\right)$ | $\tilde{\mathcal{O}}\left(\sqrt{K^2 T}\right)$ |

Table 2: Regret bounds for different algorithms in matrix games. $K$ denotes the number of actions for each player, $T$ denotes the time horizon, and $\gamma_T$ in the bound for the GP-MW algorithm denotes a kernel-dependent quantity. In this table, we assume both players have the same number of strategies. This can be generalised to the case where both players have different numbers of strategies. For the regret bound of COEBL, we consider the worst-case scenario (i.e., the opponent uses the best-response strategy) and the Nash regret (Def. (3)), the same as in (O'Donoghue et al., 2021).

## A.2 ALGORITHM IMPLEMENTATION

Previous works, including (O'Donoghue et al., 2021; Cai et al., 2023), have not released the source code for their algorithms. Therefore, we provide our own implementation for COEBL, and other bandit baselines used in this paper. The source code is available at the anonymous link `https://anonymous.4open.science/r/ICLR2025_Code-BD87/README.md`.

We will release the code later once the paper is accepted.

## A.3 PSEUDOCODE OF ALGORITHMS

As follows, we summarise a general framework of algorithms for matrix games with bandit feedback considered in this paper. We only present the algorithm for the row player, and the algorithm for the column player is symmetric.

---

**Algorithm 2** General framework for matrix games with bandit feedback (O'Donoghue et al., 2021)

---

**Require:** Policy space of player: $\mathcal{X} \subseteq \Delta_m$;
**Require:** Initial probability distribution $P_1 \in \mathcal{X}$;
1: **for** $t = 1$ to $T$ **do**
2:    The row player chooses action $i_t$ from $P_t$
3:    The column player chooses action $j_t$ from $Q_t$
4:    Observe reward $r_t$ based on $i_t, j_t$
5:    Update probability distribution $P_t$ based on $\mathcal{F}_{t+1}$, where $\mathcal{F}_{t+1} := (i_1, j_1, r_1 \cdots, i_t, j_t, r_t)$
6: **end for**

---

**Algorithm 5** EXP3-IX variant for matrix games (Cai et al., 2023)

---

**Require:** Define $\eta_t = t^{-k_\eta}$, $\beta_t = t^{-k_\beta}$, $\varepsilon_t = t^{-k_\varepsilon}$ where $k_\eta = \frac{5}{8}$, $k_\beta = \frac{3}{8}$, $k_\varepsilon = \frac{1}{8}$. $\mathcal{A}$ is the set of actions.
**Require:** $\Omega_t = \{x \in \Delta_m : x_a \geq \frac{1}{mt^2}, \forall a \in \mathcal{A}\}$.
1: **Initialisation:** $x_1 = \frac{1}{m}(1, \cdots, 1)$.
2: **for** $t = 1, 2, \ldots$ **do**
3:    Sample $a_t \sim x_t$, and receive $\sigma_t \in [0, 1]$ with $\sigma_t = G_{a_t, b_t}$ where $b_t$ is the action by the opponent.
4:    Compute $g_t$ where $g_{t,a} = 1[a_t = a]\sigma_t/(x_{t,a} + \beta_t) + \varepsilon_t \ln x_{t,a}, \forall a \in \mathcal{A}$.
5:    Update $x_{t+1} = \arg\min_{x \in \Omega_t} \left\{ x^\top g_t + \frac{1}{\eta_t} \mathrm{KL}(x, x_t) \right\}$.
6: **end for**

---

---

**Algorithm 3** EXP3 for matrix games (Auer et al., 1995; O'Donoghue et al., 2021)

---

1: **Input:** Number of actions $K$, number of iterations $T$, learning rate $\eta$ and exploration parameter $\gamma$.
2: **Initialise:**
3:   $\hat{S}_{0,i} \leftarrow 0$ for all $i \in [K]$
4: **for** $t = 1, 2, \cdots, T$ **do**
5:   Calculate the sampling distribution $P_t$: for all $i$
6:     $P_{ti} \leftarrow (1 - \gamma)\exp(\eta\hat{S}_{t-1,i})/\sum_{j=1}^{K}\exp(\eta\hat{S}_{t-1,j}) + \gamma/K$
7:   Sample $A_t \sim P_t$ and observe reward $X_t \in [0, 1]$
8:   Update $\hat{S}_{ti}$: for all $i$
9:     $\hat{S}_{ti} \leftarrow \hat{S}_{t-1,i} + 1 - 1\{A_t = i\}(1 - X_t)/P_{ti}$
10: **end for**

---

**Algorithm 4** UCB for matrix games (O'Donoghue et al., 2021)

---

1: **for** round $t = 1, 2, \ldots, T$ **do**
2:   **for** all $i, j \in [m]$ **do**
3:     compute $\tilde{A}_{ij}^t = \bar{A}_{ij}^t + \sqrt{\frac{2\log(2T^2m^2)}{1 \vee n_{ij}^t}}$
4:   **end for**
5:   use policy $x \in \arg\max_{x \in \Delta_m} \min_{y \in \Delta_m} y^T \tilde{A}^t x$
6: **end for**

---

## A.4 TECHNICAL LEMMAS

**Lemma 3.** *Given $x, y \in \Delta_m$, for all $i, j \in [m]$, $A_{ij} \in \mathbb{R}$, then $y^T A x = \sum_{i,j \in [m]} y_j x_i A_{ij}$.*

*Proof of Lemma 3.* We compute $y^T A x$ as follows.

$$y^T A x = (y_1 \quad \ldots \quad y_m) \begin{pmatrix} A_{11} & \ldots & A_{1m} \\ \vdots & \ddots & \vdots \\ A_{m1} & \ldots & A_{mm} \end{pmatrix} \begin{pmatrix} x_1 \\ \vdots \\ x_m \end{pmatrix}$$

Note that simple algebra gives

$$= (y_1 \quad \ldots \quad y_m) \begin{pmatrix} A_{11}x_1 + A_{1m}x_m \\ \vdots \\ A_{m1}x_1 + A_{mm}x_m \end{pmatrix}$$

$$= \sum_{j=1}^{m} y_j \left( \sum_{i=1}^{m} x_i A_{ij} \right)$$

$$= \sum_{i,j \in [m]} y_j x_i A_{ij}.$$

$\square$

**Lemma 4.** *The following inequalities hold for any $n \in \mathbb{N}$:*

*(1)*

$$1 + \frac{1}{\sqrt{2}} + \cdots + \frac{1}{\sqrt{n}} \leq 2\sqrt{n}.$$

*(2) Given $x_i \geq 0$ for all $i \in [n]$,*

$$\frac{1}{n} \sum_{i=1}^{n} x_i \leq \sqrt{\frac{\sum_{i}^{n} x_i^2}{n}}$$

*(3) Hoeffding's inequality for $\sigma^2$-sub-Gaussian random variables with zero-mean ([Vershynin, 2018]): let $X_1, \ldots, X_n$ be $n$ independent random variables such that $X_i$ is $\sigma^2$-sub-Gaussian. Then for any $\mathbf{a} \in \mathbb{R}^n$, we have*

$$\Pr\left(\sum_{i=1}^n a_i X_i > t\right) \le \exp\left(-\frac{t^2}{2\sigma^2\|\mathbf{a}\|_2^2}\right), \Pr\left(\sum_{i=1}^n a_i X_i < -t\right) \le \exp\left(-\frac{t^2}{2\sigma^2\|\mathbf{a}\|_2^2}\right).$$

*Of special interest is the case where $a_i = 1/n$ for all $i$. Then, we get that the average $\bar{X} = \frac{1}{n}\sum_{i=1}^n X_i$ satisfies*

$$\Pr(\bar{X} > t) \le \exp\left(-\frac{nt^2}{2\sigma^2}\right), \Pr(\bar{X} < -t) \le \exp\left(-\frac{nt^2}{2\sigma^2}\right).$$

*Proof of Lemma 4.* Proof of (3) can be found in ([Vershynin, 2018]). So, we only provide the proofs of other two inequalities here.

(1) We proceed by induction. For $n = 1$, the inequality is trivial, i.e. $1 \le 2\sqrt{1}$. Now, assume the inequality holds for $n = k \ge 2$. For the case $n = k + 1$, applying the induction hypothesis step gives,

$$1 + \frac{1}{\sqrt{2}} + \cdots + \frac{1}{\sqrt{k}} + \frac{1}{\sqrt{k+1}} = 2\sqrt{k} + \frac{1}{\sqrt{k+1}}$$

Rearranging the terms gives

$$\le 2\sqrt{k+1} + 2\sqrt{k} - 2\sqrt{k+1} + \frac{1}{\sqrt{k+1}}$$

Notice that $2\sqrt{k} - 2\sqrt{k+1} = \frac{-2}{\sqrt{k}+\sqrt{k+1}}$. Thus, we have

$$= 2\sqrt{k+1} + \frac{\sqrt{k} + \sqrt{k+1} - 2\sqrt{k+1}}{\sqrt{k+1}\left(\sqrt{k+1} + \sqrt{k}\right)}$$

Note that $\sqrt{k} + \sqrt{k+1} - 2\sqrt{k+1} = \sqrt{k} - \sqrt{k+1} < 0$ gives

$$< 2\sqrt{k+1}.$$

Thus, we complete the induction step and can complete the proof i.e. the inequality holds for all $n \in \mathbb{N}$.

(2) We proceed by induction. For $n = 1$, the inequality is trivial, i.e. $1 \le \sqrt{1^2}$. Now, assume the inequality holds for $n = k \ge 2$. For the case $n = k + 1$, applying the induction hypothesis step gives,

$$\frac{1}{(k+1)^2}\left(x_1 + \cdots + x_k + x_{k+1}\right)^2 \le \frac{1}{(k+1)^2}\left(\sqrt{k\sum_{i=1}^k x_i^2} + x_{k+1}\right)^2$$

$$= \frac{1}{(k+1)^2}\left(k\sum_{i=1}^k x_i^2 + x_{k+1}^2 + 2x_{k+1}\sqrt{k\sum_{i=1}^k x_i^2}\right)$$

Notice that $2ab \le a^2 + b^2$ for $a, b \ge 0$ gives $2x_{k+1}\sqrt{k\sum_{i=1}^k x_i^2} = 2x_{k+1}\sqrt{k} \cdot \sqrt{\sum_{i=1}^k x_i^2} \le kx_{k+1}^2 + \sum_{i=1}^k x_i^2$.

$$\le \frac{1}{(k+1)^2}\left(k\sum_{i=1}^k x_i^2 + x_{k+1}^2 + kx_{k+1}^2 + \sum_{i=1}^k x_i^2\right)$$

Rearranging the terms gives

$$= \frac{1}{(k+1)^2} \left( (k+1) \sum_{i=1}^{k+1} x_i^2 \right) = \frac{1}{k+1} \sum_{i=1}^{k+1} x_i^2.$$

Then, taking the square root of both sides gives the desired inequality for the case $n = k+1$. Thus, we complete the proof.

$\square$

### A.5 OMITTED PROOFS

Note that we restrict $A \in [0,1]^{m \times m}$ in the analysis for simplification. However, the proof works for any bounded $A \in [-b,b]^{m \times m}$ where $b$ is constant with respect to $T$ and $m$, by simply shifting from $[-b,b]$ to $[0,2b]$ and normalising the entries in $[0,1]$.

**Lemma 1.** *Suppose Assumption (A) holds with $T \geq 2m^2 \geq 2$ and $\delta := \left(1/2T^2m^2\right)^{c/8}$ where $c > 0$ is the mutation rate in COEBL. For each iteration $t \in \mathbb{N}$, given $\tilde{A}^t$ in Algorithm 1, we have:*

$$\Pr\left(A_{ij} - (\tilde{A}_t)_{ij} \leq 0\right) \geq 1 - \delta, \quad \text{for all } i,j \in [m]. \tag{2}$$

*Proof of Lemma 1.* We consider the mutation rate $c > 0$ in COEBL, where $c$ is a constant with respect to $T$ and $m$. We denote the empirical mean of the sample payoff $A_{ij}$ by $(\bar{A}_t)_{ij}$ and the number of times that row $i$ and column $j$ have been chosen by both players up to round $t$. Under Assumption (A), we compute the probability with $z_{ij} \sim \mathcal{N}(0,1)$ are i.i.d:

$$\Pr\left(A_{ij} \leq (\tilde{A}_t)_{ij}\right)$$

$$= \Pr\left( (A_{ij} \leq (\bar{A}_t)_{ij} + \sqrt{\frac{c\log(2T^2m^2)}{1 \vee n_{ij}^t + 1}} + \frac{1}{1 \vee n_{ij}^t} \cdot z_{ij} \right)$$

$$= \Pr\left( A_{ij} - \frac{1}{1 \vee n_{ij}^t} \sum_{k=1}^{1 \vee n_{ij}^t} (A_k)_{ij} - \frac{z_{ij}}{1 \vee n_{ij}^t} \leq \sqrt{\frac{c\log(2T^2m^2)}{1 \vee n_{ij}^t + 1}} \right)$$

Recall that $(A_k)_{ij} = A_{ij} + \eta_k$ where $\eta_k$ are i.i.d. 1-sub-Gaussian with zero mean. Note that $\eta_k' := -\eta_k$ is also 1-sub-Gaussian with zero mean and $z_{ij}' := -z_{ij} \sim \mathcal{N}(0,1)$. Thus, we can rewrite the inequality as follows.

$$= \Pr\left( \frac{1}{1 \vee n_{ij}^t} \left( \sum_{k=1}^{1 \vee n_{ij}^t} \eta_k' + z_{ij}' \right) \leq \sqrt{\frac{c\log(2T^2m^2)}{1 \vee n_{ij}^t + 1}} \right)$$

We consider the reverse quantity:

$$\Pr\left( \frac{1}{1 \vee n_{ij}^t} \left( \sum_{k=1}^{1 \vee n_{ij}^t} \eta_k' + z_{ij}' \right) > \sqrt{\frac{c\log(2T^2m^2)}{1 \vee n_{ij}^t + 1}} \right)$$

$$= \Pr\left( \frac{1}{1 \vee n_{ij}^t + 1} \left( \sum_{k=1}^{1 \vee n_{ij}^t} \eta_k' + z_{ij}' \right) > \frac{1 \vee n_{ij}^t}{1 \vee n_{ij}^t + 1} \sqrt{\frac{c\log(2T^2m^2)}{1 \vee n_{ij}^t + 1}} \right)$$

Note that $\frac{1 \vee n_{ij}^t}{1 \vee n_{ij}^t + 1} \geq \frac{1}{2}$. Thus, we have

$$\leq \Pr\left( \frac{1}{1 \vee n_{ij}^t + 1} \left( \sum_{k=1}^{1 \vee n_{ij}^t} \eta_k' + z_{ij}' \right) > \frac{1}{2} \sqrt{\frac{c\log(2T^2m^2)}{1 \vee n_{ij}^t + 1}} \right)$$

Using Hoeffding's inequality for i.i.d. sub-Gaussian random variables gives

$$\leq \exp\left(-\frac{(1 \vee n_{ij}^t + 1) \cdot \frac{1}{4}\frac{c\log(2T^2m^2)}{1\vee n_{ij}^t+1}}{2}\right)$$

$$= \left(\frac{1}{2T^2m^2}\right)^{c/8} := \delta$$

Hence, we complete the proof. □

**Theorem 2** (Main Result). *Consider any two-player zero-sum matrix game. Under Assumption (A) with $T \geq 2m^2 \geq 2$ and $\delta = \left(1/2T^2m^2\right)^{c/8}$, where $c > 0$ is the mutation rate in COEBL, the worst-case Nash regret of COEBL for $c \geq 8$ is bounded by $2\sqrt{2cTm^2\log(2T^2m^2)}$, i.e., $\tilde{O}(\sqrt{m^2T})$.*

*Proof of Theorem 2.* First, we follow the proof of Theorem 1 in (O'Donoghue et al., 2021) using the following events. Let $E_t$ be the event that $\exists i, j \in [m]$ such that $(\tilde{A}_t)_{ij} < A_{ij}$. We know $E_t \in \mathcal{F}_t$ where $\mathcal{F}_t$ is defined in the preliminaries. Consider some iteration $E_t$ does not hold and let $\tilde{y}_t := \operatorname{argmin}_{y\in\Delta_m} y_t^T\tilde{A}_t x_t$ be the best response of the column player. Since $E_t$ does not hold, then for $\forall i,j \in [m]$, $A_{ij} \leq (\tilde{A}_t)_{ij}$. Thus, $V_A^* \leq V_{\tilde{A}_t}^*$. So, the regret in each round $t$ under the case that $E_t$ does not hold is bounded by the following,

$$V_A^* - \mathrm{E}_t\left(y_t^T A x_t\right) \leq \mathrm{E}_t\left(V_{\tilde{A}_t}^* - y_t^T A x_t\right) = \mathrm{E}_t\left(\tilde{y}_t^T \tilde{A}_t x_t - y_t^T A x_t\right)$$

Recall that $\tilde{y}_t$ is the best response of the column player.

$$\leq \mathrm{E}_t\left(y_t^T \tilde{A}_t x_t - y_t^T A x_t\right)$$

$$= \mathrm{E}_t\left(y_t^T\left(\tilde{A}_t - A\right)x_t\right)$$

Recall the estimated matrix in Algorithm 1. We have $\left(\tilde{A}_t - A\right)_{ij} = \sqrt{\frac{c\log(2T^2m^2)}{1\vee n_{ij}^t+1}} + \frac{1}{1\vee n_{ij}^t}\mathcal{N}(0,1)$. Note that $\log(2T^2m^2) = \log\left((1/\delta)^{8/c}\right) = 8\log(1/\delta)/c$. Using Lemma 3 gives

$$= \mathrm{E}_t\left(\sqrt{\frac{8\log(1/\delta)}{1\vee n_{i_tj_t}^t+1}} + \sum_{j=1}^m y_j \sum_{i=1}^m x_i \frac{z_{ij}}{1\vee n_{ij}^t}\right)$$

Note that $1 \vee n_{ij}^t \geq 1$. We can have the following inequality.

$$\leq \mathrm{E}_t\left(\sqrt{\frac{8\log(1/\delta)}{1\vee n_{i_tj_t}^t+1}} + \sum_{j=1}^m y_j \sum_{i=1}^m x_i z_{ij}\right)$$

By linearity of expectation and $\mathrm{E}_t(z_{ij}) = 0$, we have

$$= \mathrm{E}_t\left(\sqrt{\frac{8\log(1/\delta)}{1\vee n_{i_tj_t}^t+1}}\right). \tag{4}$$

Thus, we can bound the overall regret. Given the class of games $\forall A \in \mathcal{A}$ defined in (A), we have

$$\mathcal{R}\left(A, \text{COEBL}, T\right) = \mathrm{E}\left(\sum_{t=1}^T V_{A^*} - \mathrm{E}_t\left(y_t^T A x_t\right)\right)$$

Using law of total probability gives

$$= \mathrm{E}\left(\sum_{t=1}^T V_{A^*} - \mathrm{E}_t\left(y_t^T A x_t\right) \mid \cap_{t=1}^T E_t^c\right) \cdot \mathrm{Pr}\left(\cap_{t=1}^T E_t^c\right)$$

$$+ \mathrm{E}\left(\sum_{t=1}^T V_{A^*} - \mathrm{E}_t\left(y_t^T A x_t\right) \mid \left(\cap_{t=1}^T E_t^c\right)^c\right) \times \mathrm{Pr}\left(\left(\cap_{t=1}^T E_t^c\right)^c\right)$$

Using De Morgan's Law gives $\left(\cap_{t=1}^{T} E_t^c\right)^c = \cup_{t=1}^{T} E_t$.

$$= \mathrm{E}\left(\sum_{t=1}^{T} V_{A^*} - \mathrm{E}_t\left(y_t^T A x_t\right) \mid \cap_{t=1}^{T} E_t^c\right) \cdot \mathrm{Pr}\left(\cap_{t=1}^{T} E_t^c\right)$$

$$+ \mathrm{E}\left(\sum_{t=1}^{T} V_{A^*} - \mathrm{E}_t\left(y_t^T A x_t\right) \mid \cup_{t=1}^{T} E_t\right) \cdot \mathrm{Pr}\left(\cup_{t=1}^{T} E_t\right)$$

Using the upper bound in Eq. 4 and $\mathrm{Pr}\left(\cap_{t=1}^{T} E_t^c\right) \le 1$ gives

$$\le \mathrm{E}\left(\sum_{t=1}^{T} \sqrt{\frac{8\log(1/\delta)}{1 \vee n_{i_t j_t}^t + 1}}\right) + \mathrm{E}\left(\sum_{t=1}^{T} 1\right) \cdot \mathrm{Pr}\left(\cup_{t=1}^{T} E_t\right)$$

Using the Union bound gives

$$\le \mathrm{E}\left(\sum_{t=1}^{T} \sqrt{\frac{8\log(1/\delta)}{1 \vee n_{i_t j_t}^t + 1}}\right) + T \sum_{t=1}^{T} \mathrm{Pr}\left(E_t\right)$$

Using Lemma 1 gives $\mathrm{Pr}\left(E_t\right) \le \delta$. Thus, we have

$$\le \mathrm{E}\left(\sum_{t=1}^{T} \sqrt{\frac{8\log(1/\delta)}{1 \vee n_{i_t j_t}^t + 1}}\right) + T^2 \delta$$

Recall that $\delta = \left(1/2T^2 m^2\right)^{c/8} \le 1/2T^2 m^2$ for $c \ge 8$. Note that $\log(1/\delta) = \log\left(\left(2T^2 m^2\right)^{c/8}\right) = c\log(2T^2 m^2)/8$.

$$\le \mathrm{E}\left(\sum_{t=1}^{T} \sqrt{\frac{c\log(2T^2 m^2)}{1 \vee n_{i_t j_t}^t + 1}}\right) + \frac{1}{2m^2}$$

Rewrite the summation in the expectation.

$$\le \sum_{i,j\in[m]} \mathrm{E}\left(\sum_{t=1}^{T} \sqrt{\frac{c\log(2T^2 m^2)}{1 \vee n_{ij}^t + 1}} \mathbb{1}_{\{i_t = i, j_t = j\}}\right) + \frac{1}{2m^2}$$

Let us denote the set $B_{ij} := \{t \in \{0, \cdots, n_{ij}^T\} \mid i_t = i, j_t = j\}$ for $i, j \in [m]$. So we can rewrite the summand as follows.

$$= \sqrt{c\log(2T^2 m^2)} \sum_{i,j\in[m]} \mathrm{E}\left(\sum_{t_k \in B_{ij}} \sqrt{\frac{1}{1 \vee n_{ij}^{t_k} + 1}}\right) + \frac{1}{2m^2}$$

Note that $n_{ij}^{t_k}$ is an increasing sequence in $t_k$. Thus, we can have

$$= \sqrt{c\log(2T^2 m^2)} \sum_{i,j\in[m]} \mathrm{E}\left(\sum_{k=1}^{n_{ij}^T} \sqrt{\frac{1}{k+1}}\right) + \frac{1}{2m^2}$$

Adding one more $1/\sqrt{1}$ in the inner sum and using Lemma 4 (1) give

$$\le \sqrt{c\log(2T^2 m^2)} \sum_{i,j\in[m]} \mathrm{E}\left(2\sqrt{1 \vee n_{ij}^T + 1}\right) + \frac{1}{2m^2}$$

Using Lemma 4 (2) with $x_k := \sqrt{1 \vee n_{ij}^T + 1}$ where $k \in [m^2]$ gives

$$\le \sqrt{4c\log(2T^2 m^2)} \cdot m^2 \sqrt{\frac{\sum_{i,j\in[m]} 1 \vee n_{ij}^T + 1}{m^2}}$$

Notice that $1 \vee n_{ij}^T \leq n_{ij}^T + 1$.

$$\leq \sqrt{4c \log(2T^2 m^2)} \sqrt{m^2 \sum_{i,j \in [m]} (n_{ij}^T + 2)}$$

Notice that $\sum_{i,j \in [m]} n_{ij}^T = T$.

$$= \sqrt{4c \log(2T^2 m^2)} \sqrt{m^2 (T + 2m^2)}$$

Since $T \geq 2m^2$, we have

$$\leq \sqrt{4c \log(2T^2 m^2)} \sqrt{2m^2 T} = 2\sqrt{2cTm^2 \log(2T^2 m^2)} = \tilde{O}(\sqrt{m^2 T}).$$

Thus, we can conclude that $\text{WORSTCASEREGRET}(\mathcal{A}, \text{COEBL}, T) = \tilde{\mathcal{O}}\left(\sqrt{m^2 T}\right)$. $\qquad\square$

### A.6 COMPLETE EMPIRICAL RESULTS

#### A.6.1 REASONS FOR THE CHOICES OF MATRIX GAMES BENCHMARKS

We choose the given matrix games benchmarks for the following reasons:

1. The RPS game is a classical benchmark widely used in the previous RL and game theory literature, and we want to compare the performance of COEBL with the existing algorithms.

2. However, RPS consists of a small number of actions and the game is not complex enough to test the performances of the algorithms. Therefore, we included the DIAGONAL and BIGGERNUMBER games, which are more complex and feature exponentially larger action spaces

3. We chose these matrix game benchmarks from multiple fields, including RL (Littman, 1994; O'Donoghue et al., 2021), game theory (Zhang & Sandholm, 2024), and evolutionary computation (Lehre & Lin, 2024), to demonstrate the general applicability of the proposed algorithm.

#### A.6.2 REASONS FOR THE CHOICES OF SYMMETRIC MATRIX GAMES BENCHMARKS

One might notice that all the matrix games considered in the experiments are symmetric, meaning that for the payoff matrix $A$, $A_{ij} = -A_{ji}$ for all $i, j \in [m]$. In such games, there is no advantage in being the first or second player, the experimental studies provide fair head-to-head comparisons.

#### A.6.3 DIAGONAL GAME

We defer the full experimental results on DIAGONAL game to the appendix and provide the payoff matrix of DIAGONAL game when $n = 2$.

|    | 00 | 01 | 10 | 11 |
|----|----|----|----|----|
| 00 | 0  | -1 | -1 | -1 |
| 01 | 1  | 0  | 0  | -1 |
| 10 | 1  | 0  | 0  | -1 |
| 11 | 1  | 1  | 1  | 0  |

Table 3: The payoff matrix of DIAGONAL game ($n = 2$). Binary bitstrings represent different pure strategies of each player. This game compares the number of 1-bits of each player.

In this case, both players have $2^n$ actions, which is way more complicated than the RPS. In terms of the regret, all the algorithms in the self-play scenario, exhibit sublinear regrets. However, only COEBL converges for several problem instances. When $n$ increases to certain level, like $n \geq 4$, none of them can converge to the Nash equilibrium anymore. For the ALG-1 vs ALG-2 scenario, after iteration 2000, COEBL has an overwhelming advantage over other bandit baselines in terms of regret performance. For the convergence of the the Nash equilibrium, surprisingly, in Figure 8, UCB-vs-COEBL converges to or approximates the Nash equilibrium even when $n = 4$. However, they also fail to converge to the Nash equilibrium when $n = 5, 6, 7$. We can see that the opponent performance has certain impact to the overall dynamics towards the Nash equilibrium.

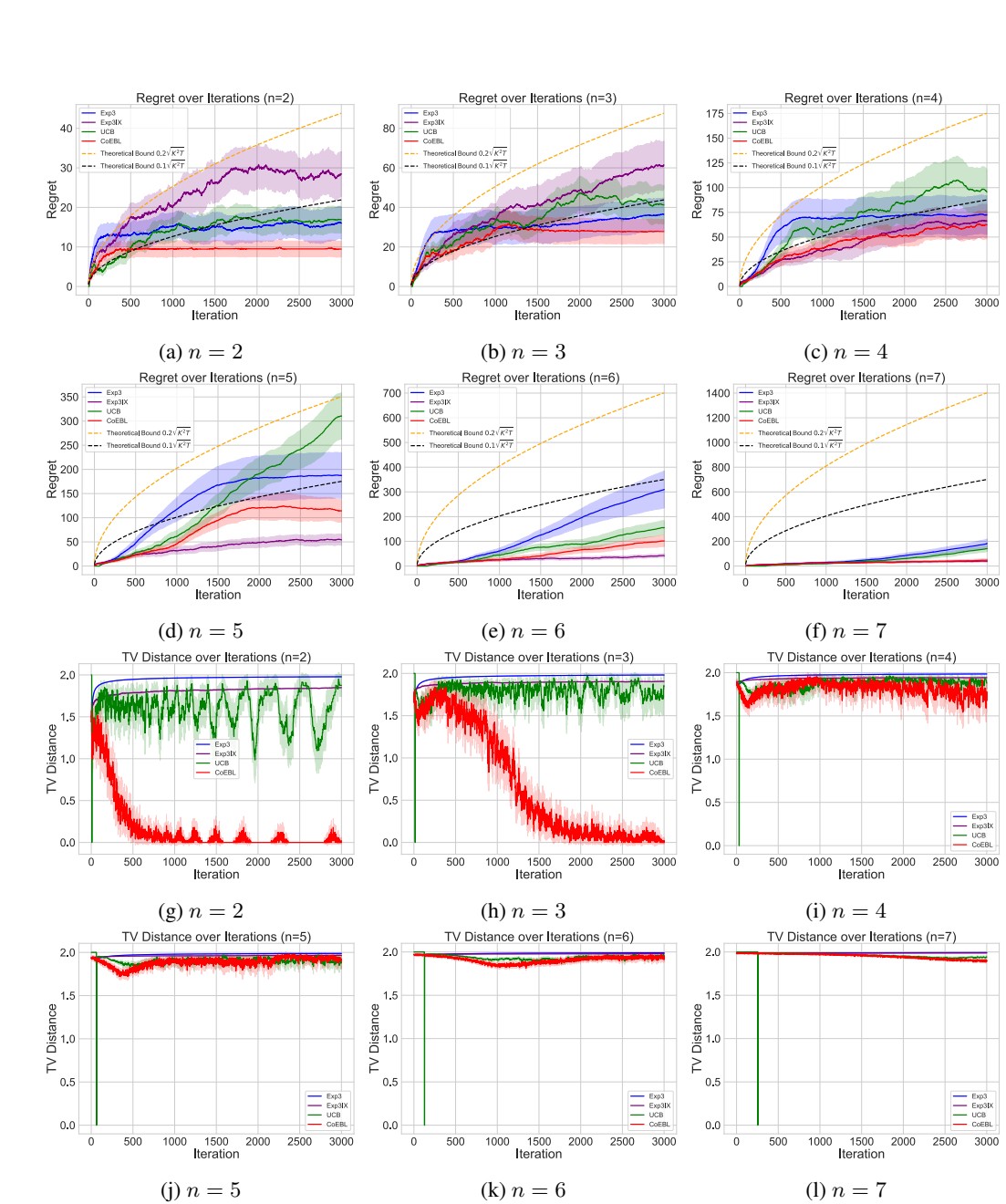

Figure 7: Regret and TV Distance for Self-Plays on DIAGONAL for $n = 2, \ldots, 7$

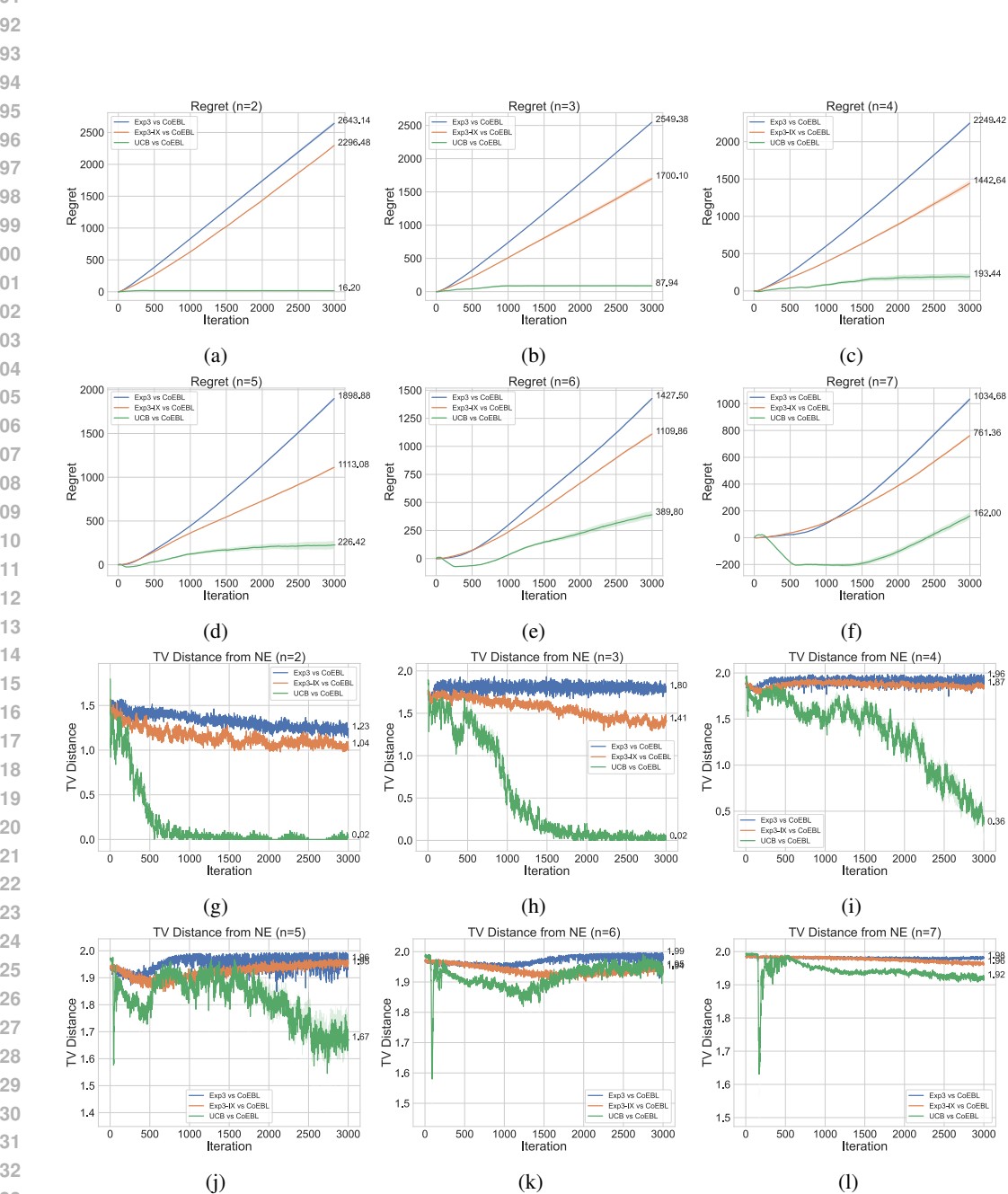

Figure 8: Regret and TV-distance for ALG 1-vs-ALG 2 on DIAGONAL for $n = 2, \ldots, 7$.

### A.6.4 BIGGERNUMBER GAME

BIGGERNUMBER is another challenging two-player zero-sum game proposed and analysed by Zhang & Sandholm (2024). In this game, each player aims to select a number greater than their opponent's. The players' action space is defined as $\mathcal{X} = \{0, 1\}^n$, where binary bitstrings of length $n$ correspond to natural numbers in the range $[0, 2^n - 1]$. If the players select the same number, they receive a payoff of 0. If the difference between the players' numbers is exactly 1, the player with the larger number receives a payoff of 2, while the player with the smaller number receives $-2$. Otherwise, the player with the larger number receives a payoff of 1, and the player with the smaller number receives $-1$. To simplify the game and align it with ternary games, we modify the payoff function BIGGERNUMBER : $\mathcal{X} \times \mathcal{X} \to \{-1, 0, 1\}$ defined by:

$$\text{BIGGERNUMBER}(x, y) := \begin{cases} 0 & x = y \\ 1 & x > y \\ -1 & x < y \end{cases}.$$

The payoff matrix of the BIGGERNUMBER game for $n = 2$ is:

|    | 00 | 01 | 10 | 11 |
|----|----|----|----|----|
| 00 | 0  | -1 | -1 | -1 |
| 01 | 1  | 0  | -1 | -1 |
| 10 | 1  | 1  | 0  | -1 |
| 11 | 1  | 1  | 1  | 0  |

Table 4: The payoff matrix of the BIGGERNUMBER game for $n = 2$. Binary bitstrings represent the pure strategies available to each player: $0 = (00)_2$, $1 = (01)_2$, $2 = (10)_2$, and $3 = (11)_2$. In this game, players compare their numbers from $\mathbb{N}$.

As proved by Zhang & Sandholm (2024), this payoff matrix also exhibits a unique pure Nash equilibrium where both players choose $1^n \in \{0, 1\}^n$ (i.e., the binary string of all ones, corresponding to $2^n - 1 \in \mathbb{N}$). This corresponds to the mixed Nash equilibrium $x^* = (0, \cdots, 1)$ and $y^* = (0, \cdots, 1)$. We conduct experiments using Algorithms 3 to 5 and compare them with our proposed Algorithm 1 (i.e. COEBL) on this matrix game benchmark, the BIGGERNUMBER game.

In Figure 9, we present the self-play results of each algorithm on the BIGGERNUMBER game for various values of $n$. We observe that COEBL exhibits sublinear regret in the BIGGERNUMBER game, similar to other bandit baselines, and aligns with our theoretical bound. In terms of convergence measured by TV-distance, COEBL converges to the Nash equilibrium for $n = 2, 3, 4$, while the other baselines do not converge. However, after $n = 5$ (as the number of pure strategies increases exponentially), COEBL also fails to converge to the Nash equilibrium.

In Figure 10, we present the regret and TV-distance for ALG 1-vs-ALG 2 on BIGGERNUMBER. Similar to the DIAGONAL game, we observe that all regret values are positive with minimum 8.39 and maximum 351.27, indicating that the minimiser is winning on average. Thus, COEBL outperforms the other bandit baselines in BIGGERNUMBER for all $n = 2, \ldots, 7$.

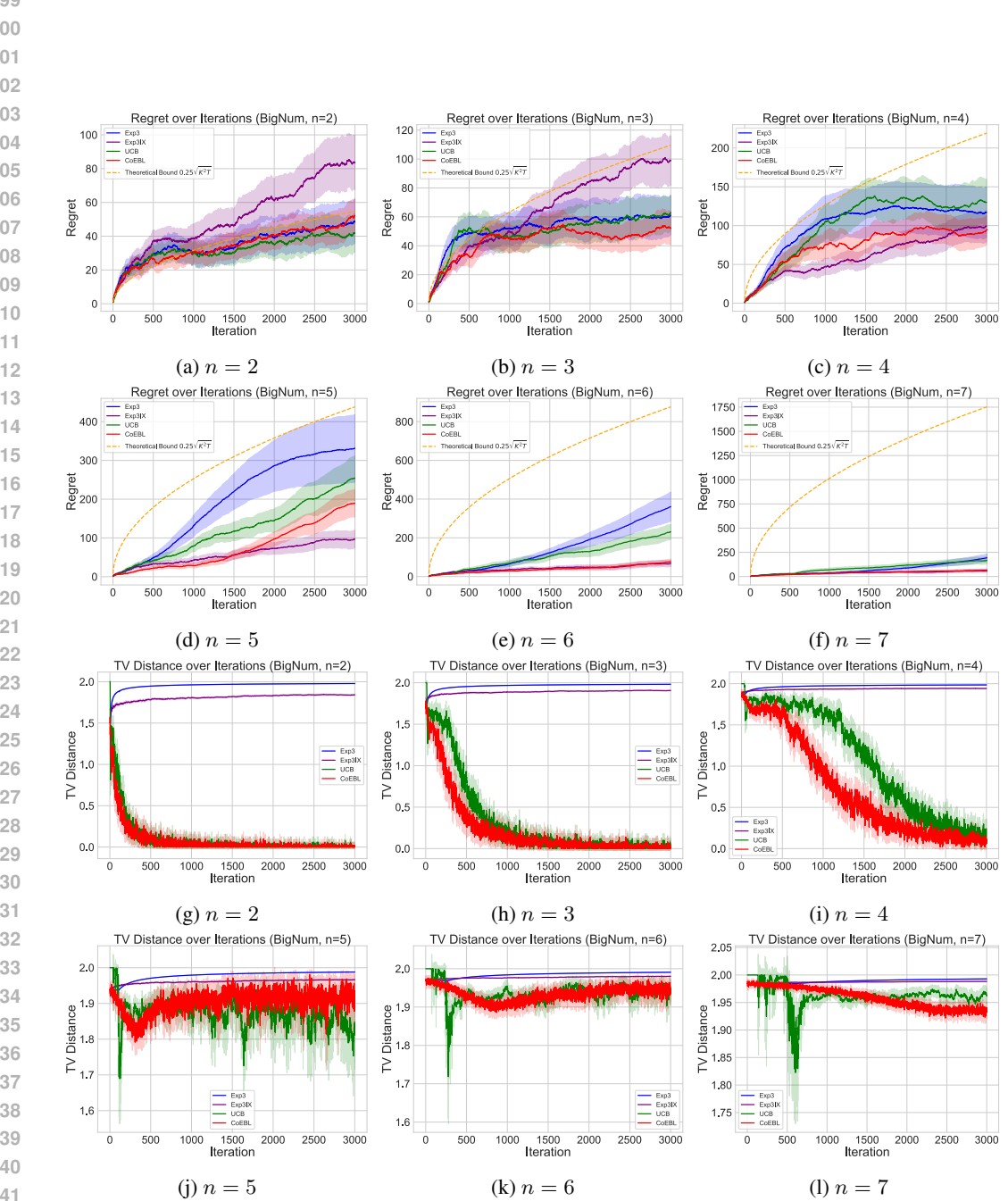

Figure 9: Regret and TV Distance for Self-Plays on BIGGERNUMBER

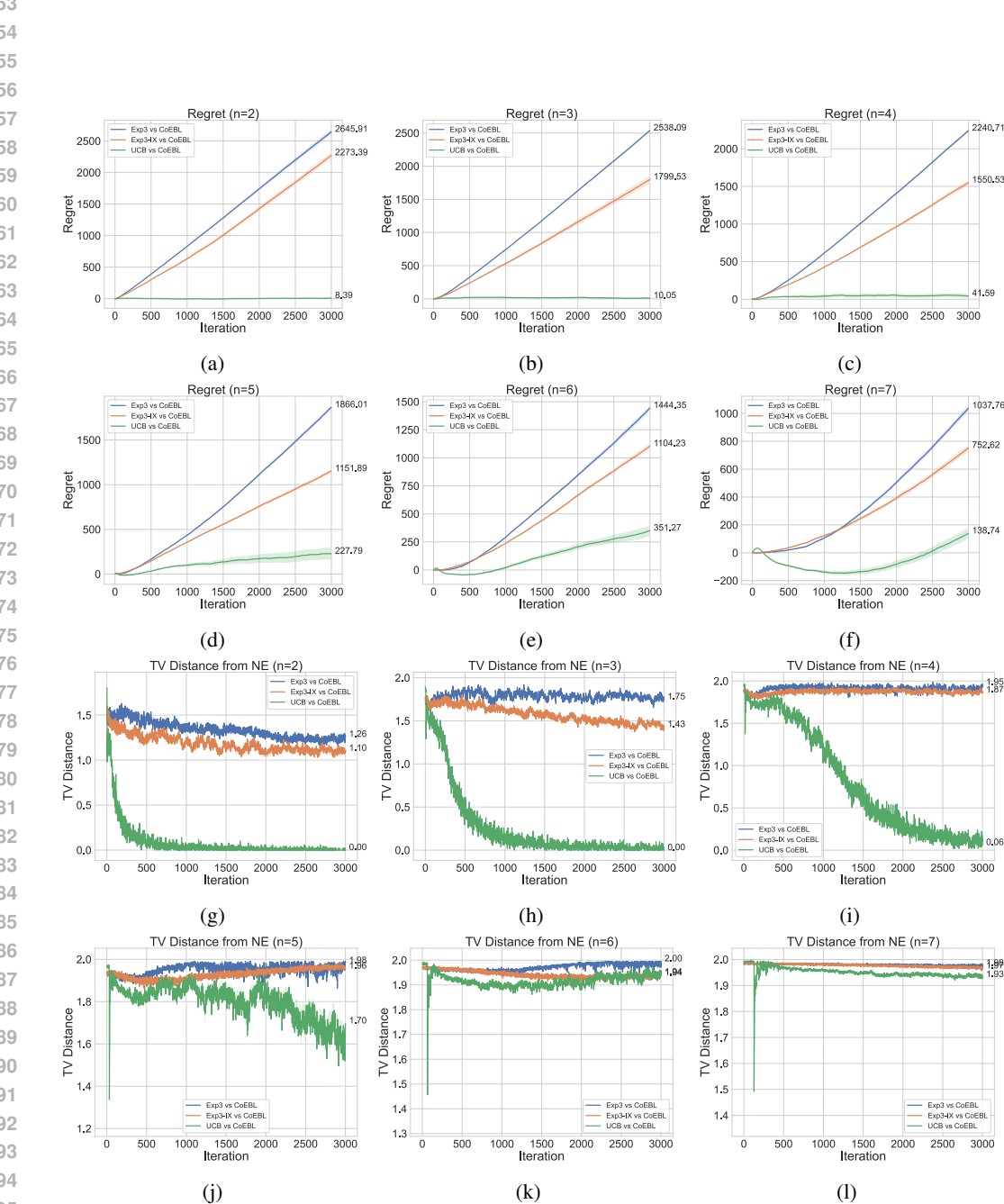

Figure 10: Regret and TV-distance for ALG 1-vs-ALG 2 on BIGGERNUMBER.