# OpenReview forum: "Competitive Co-Evolutionary Learning on Matrix Games with Bandit Feedback"
_ICLR.cc/2025/Conference — ICLR 2025 Conference Withdrawn Submission_

### Official Review · Reviewer_KadD · 2024-11-03

**Soundness:** 2
**Presentation:** 2
**Contribution:** 2
**Rating:** 5
**Confidence:** 3

**Summary:**

This paper proposes the Competitive Co-evolutionary Bandit Learning (COEBL) algorithm for two-player zero-sum matrix games with bandit feedback, introducing evolutionary algorithms (EAs) to achieve randomized optimism.  COEBL leverages EAs to handle noisy payoff data, which achieves sublinear regret while outperforming traditional bandit algorithms such as EXP3 and UCB. The authors offer a theoretical analysis of COEBL’s regret and conduct  empirical tests across multiple matrix game benchmarks, including the Rock-Paper-Scissors, Diagonal, and BiggerNumber games.

**Strengths:**

- Originality: Integrating evolutionary algorithms into bandit learning for randomized optimism is novel
- Quality: Rigorous proofs and thorough empirical evaluation enhance the paper’s credibility. The authors carefully benchmark COEBL against established algorithms across multiple scenarios, providing a comprehensive assessment.

**Weaknesses:**

- Technical clarity: some details on the evolutionary operations, e.g., mutation and selection mechanisms, could be expanded to clarify their implementation within COEBL.
- The definition of regret as well as the worst case regret might not serve as a good performance metric. The regret can fluctuate (potentially becoming negative in some rounds and positive in others), simple summation might cause values to "cancel out," potentially distorting performance insights. This can make upper bounds on cumulative regret somewhat misleading, especially if the goal is to understand how consistently an algorithm approximates optimal play. The *absolute regret* in some sense is a more reasonable performance metric, however, it is only considered in the empirical experiments rather than theoretical analysis.
- Thm 2 provide certain gurantees for self-play of COEBL, but there are no theoretical analysis for alg1 vs COEBL or COEBL vs alg2.
- Related work: the mutation step of COEBL shares similarity with the optimistic sampling method in the paper "Optimistic Thompson Sampling for No-Regret Learning in Unknown Games", which is not discussed in the paper.
- Technical novelty: the technical novelty for proving the main theoretical results (Thm 2) is not clear.

**Questions:**

- Why do the authors not consider integrate EAs into the multi-armed bandit framework first?
- The paper presents the experiments showing other algorithms vs COEBL (e.g. Fig. 4), and claims that COEBL outperforms the other
bandit algorithms based on the observation of linear regret. However, what would the plots for COEBL vs other algorithms be like? Is it possible to exhibit linear regret?

Minor issues
- Fig 1 a cut off at iter = 3000, where the regret of COEBL seems to surpass Exp3IX
- In Algorithm 1, the steps for choosing actions i_t, j_t are missing
- Why does the paper consider the square matrix, rather than more general matrix (non-square)?

---

### Official Review · Reviewer_zZvv · 2024-11-03

**Soundness:** 2
**Presentation:** 2
**Contribution:** 1
**Rating:** 3
**Confidence:** 5

**Summary:**

The paper introduces a novel algorithm called Competitive Co-Evolutionary Bandit Learning (COEBL) for learning in two-player zero-sum matrix games with bandit feedback. The key idea is integrating evolutionary algorithms into the bandit learning framework to achieve randomized optimism, addressing an open question about whether randomized optimism can achieve sublinear regret in such games. The authors provide theoretical analysis showing that COEBL achieves sublinear regret comparable to deterministic optimism approaches. They also conduct empirical evaluations demonstrating that COEBL outperforms classical bandit algorithms like EXP3, UCB, and EXP3-IX across various matrix game benchmarks.

**Strengths:**

- **Originality:** The paper proposes the novel integration of evolutionary algorithms into bandit learning for matrix games, introducing randomized optimism through evolutionary operators.
- **Theoretical Analysis:** Provides regret analysis showing that COEBL achieves sublinear regret comparable to deterministic approaches.
- **Empirical Results:** Demonstrates that COEBL outperforms classical bandit algorithms in various matrix game benchmarks.
- **Motivation:** Addresses an open question about the effectiveness of randomized optimism in achieving sublinear regret in matrix games.

**Weaknesses:**

- **Lack of Discussion with Closely Related Articles:** The paper fails to discuss recent closely related work, particularly "Optimistic Thompson Sampling for No-Regret Learning in Unknown Games" by Li et al., which also tackles learning in unknown games using bandit feedback and introduces the Optimism-then-NoRegret (OTN) framework and Optimistic Thompson Sampling (OTS) algorithms. This omission is significant, as Li et al.'s work addresses similar challenges and use stochastic optimism (OTS) to tackle game setting.
- **Limited Scope:** The analysis is restricted to two-player zero-sum matrix games, limiting the generality of the results.
- **Assumptions:** The reliance on sub-Gaussian noise assumptions may limit the applicability of the theoretical results to broader settings.
- **Incomplete Experimental Comparison:** The experiments lack comparisons with recent algorithms that address similar problems, such as those based on Thompson Sampling and the OTN framework.
- **Presentation Issues:** Some sections lack clarity, and mathematical notations are inconsistent. Figures could be better explained.
- **Insufficient Understanding of Stochastic Optimism and Selection Scheme**

**Questions:**

1. How does COEBL compare to the Optimistic Thompson Sampling (OTS) algorithms introduced by Li et al. in terms of theoretical guarantees and empirical performance? How the stochastic optimism in COEBL differs from that in OTS?
2. Can the COEBL framework be extended to multi-player or general-sum games? If so, what challenges would need to be addressed?
3. What is the impact of different evolutionary operators on the performance of COEBL? Have other operators been considered?
4. How sensitive is COEBL's performance to the sub-Gaussian noise assumption? Can the analysis be extended to weaker noise conditions?
5. What is the computational complexity of COEBL compared to related works?
6. Can you provide any rigorous understanding of the stochastic optimism (mutation/ optimism thompson sampling) compared to deterministic optimism?  Could you provide ablation analysis of how stochastic optimism in COEBL differs from deterministic optimism in terms of exploration-exploitation trade-off, or to discuss any theoretical insights into why stochastic optimism might be more effective in certain game settings.
7. What is the real benefit about the selection scheme. How the selection scheme in COEBL contributes to its performance compared to other algorithms. Any ablation study to isolate the impact of the selection scheme.

---

### Official Review · Reviewer_XVnb · 2024-11-04

**Soundness:** 2
**Presentation:** 2
**Contribution:** 1
**Rating:** 5
**Confidence:** 4

**Summary:**

This paper studies learning in matrix games with bandit feedback. It purposes a new algorithm that combines the UCB algorithm and evolutionary algorithms and provides some theoretical analysis of the proposed algorithm. Specifically, the paper provides a theoretical sublinear convergence guarantee of the Nash regret bound. Numerical results are provided with comparisons with other benchmark algorithms.

**Strengths:**

1. This work introduces evolutionary algorithms to the previous UCB algorithm provided in [1], allowing a stochastic version of the estimation $\tilde{A}^t$, improving the result of [1] which relies on the optimism certainty.

2. Detailed numerical results are provided under 3 different environments as long as the comparisons with other benchmark algorithms.

3. Sublinear bound of Nash regret is provided.

[1] Brendan O’Donoghue, Tor Lattimore, and Ian Osband. Matrix games with bandit feedback. In Uncertainty in Artificial Intelligence, pp. 279–289. PMLR, 2021.

**Weaknesses:**

1. The theoretical analysis of this work is based on the Nash regret $\mathbb{E}(\sum_{t = 1}^TV^*_A - r_t)$ which is not the standard regret (e.g. in [2], [3]). Specifically, the sublinear bound of Nash regret does not necessarily guarantee convergence towards the Nash equilibrium when both players deploy the same learning algorithm. This is in sharp contrast with the convergence guarantee provided in previous work [2], [3]. However, this paper does not provide a detailed literature review of the notion of Nash regret or the motivation of analyzing this regret.

2. Due to the different notion with regret, the table in section A.1 is comparing rates of different convergence guarantees. For example, [2] provides $\tilde{O}(A^{1/2}T^{-1/8})$ last iterate convergence guarantee which is different with this work in both the regret notion and convergence properties.

3. The theoretical novelty of this paper is limited, the proof of the theorem 2 follows from the same idea with the proof of theorem 1 in [1].

4. The use of selection mechanism in Algorithm 1 (line 7 - 10) is unclear. The proof of theorem 2 relies on the fact that $V^*_{\tilde{A}_t} = \tilde{y}_t^T\tilde{A}_tx_t$ which is always the case when $x_t = x'$. It is unclear why Algorithm 1 adopts this selection mechanism.

[1] Brendan O’Donoghue, Tor Lattimore, and Ian Osband. Matrix games with bandit feedback. In Uncertainty in Artificial Intelligence, pp. 279–289. PMLR, 2021.

[2] Yang Cai, Haipeng Luo, Chen-Yu Wei, and Weiqiang Zheng. Uncoupled and Convergent Learning in Two-Player Zero-Sum Markov Games with Bandit Feedback. In Thirty-seventh Conference on Neural Information Processing Systems, volume 36, 2023.

[3] Gergely Neu. Explore No More: Improved High-Probability Regret Bounds for Non-Stochastic Bandits. Advances in Neural Information Processing Systems, 28, 2015.

**Questions:**

1. The paper main focuses on ternary two-player zero-sum games. However, to me the results hold for any game satisfying assumption (A). I wonder whether if the results could be extended into richer settings beyond ternary games?

2. As discussed in Weaknesses, [2] and [3] provides theoretical guarantees for the algorithm to converge to Nash. However, in the experimental settings of this paper, there are settings where all the algorithms fail to converge to Nash. What might be the potential cause of this issue?

[2] Yang Cai, Haipeng Luo, Chen-Yu Wei, and Weiqiang Zheng. Uncoupled and Convergent Learning in Two-Player Zero-Sum Markov Games with Bandit Feedback. In Thirty-seventh Conference on Neural Information Processing Systems, volume 36, 2023.

[3] Gergely Neu. Explore No More: Improved High-Probability Regret Bounds for Non-Stochastic Bandits. Advances in Neural Information Processing Systems, 28, 2015.

---

### Official Review · Reviewer_r3P7 · 2024-11-06

**Soundness:** 2
**Presentation:** 3
**Contribution:** 2
**Rating:** 5
**Confidence:** 3

**Summary:**

The paper proposes a novel algorithm, CoEBL, based on evolutionary learning to play two-player zero-sum games with bandit feedback (unknown payoff matrix). Regret guarantees are provided, that match the ones of deterministic UCB algorithm. Experiments demonstrate that CoEBL outperforms UBC and Exp3 variants, presumably due to higher exploration.

**Strengths:**

The paper studies an interesting bandit problem and the algorithm proposed is, to the best of my knowledge, not studied in the literature. Experiments are somewhat extensive and demonstrate the superiorness of CoEBL to the other baselines.

**Weaknesses:**

The main weaknesses are as follows:
- From my understanding, CoEBL assumes (noisy) bandit feedback but it *also* requires observing the action chosen by the opponent. Thus, the feedback is more restrictive than Exp3, or am I missing something? Also, Table 2 in the Appendix should be updated accordingly.
- In Line 169, "In this paper, we mainly focus on ternary two-player zero-sum games". Does this mean the theoretical analysis holds only for such games or is this related to the experiments. It would be good to clarify it. In the first case, the statements in abstract and introduction should be adapted to refer to such a restrictive setting.
- How does CoEBL compares to Thompson sampling? My understanding is that CoEBL should coincide with the Thompson sampling variant of the approach in (O’Donoghue et al., 2021). If so, the authors should remark this and explain what are the key novel technical challenges in proving the regret bounds. If not, this should be a natural baseline in the experiments.
- The authors claim the superior performance in the experiments to be due to enhanced exploration. I think it would be good to demonstrate this more qualitatively and/or quantitatively, e.g. measuring diversity of the sampled matrices and played actions.

**Questions:**

See weaknesses.

---

### Note · Authors · 2024-11-26

**Comment:**

**1.Summary of Reviews**

We sincerely thank all reviewers for their detailed and thoughtful feedback. Below, we summarise the strengths and weaknesses of our work and address specific issues raised by reviewers.

**Strengths:**

Novelty:
CoEBL introduces evolutionary algorithms into the bandit framework, offering randomised optimism—a novel contribution addressing open questions in a game-theoretic setting.

Theoretical Contributions:
Sublinear regret guarantees comparable to deterministic approaches like UCB, tailored to noisy payoff settings.

Empirical Validation:
Experiments across diverse benchmarks (e.g., Rock-Paper-Scissors) demonstrate CoEBL's superior performance over EXP3, UCB, and EXP3-IX.

Motivation and Relevance:
The algorithm expands the applicability of evolutionary mechanisms to bandit feedback in matrix games.

**2.Weaknesses and Reviewer-Specific Issues:**

**Literature and Related Work (zZvv, KadD, r3P7, zZvv):** The omission of recent works, such as "Optimistic Thompson Sampling for No-Regret Learning in Unknown Games" (Li et al.), is significant. CoEBL must be positioned against such frameworks.
Highlight key differences between CoEBL and Thompson Sampling (TS), focusing on evolutionary mechanisms like variation and selection.


**Theoretical Analysis (XVnb, KadD, r3P7):** The use of Nash regret instead of standard regret limits convergence guarantees to Nash equilibria. We will provide a more detailed motivation for the choice of Nash regret, its significance in game-theoretic contexts, and its relationship to practical applications. Clarify whether CoEBL's analysis extends beyond ternary games and address potential limitations in broader settings.

**Experimental Comparisons (r3P7, zZvv, KadD):** Include missing baselines (e.g., TS and OTS) and explore CoEBL's relative performance in various scenarios.
Improve experimental details, including the role of evolutionary operators and sensitivity to noise assumptions.

**Technical Clarity and Presentation (r3P7, KadD, XVnb):** Improve descriptions of evolutionary operators (e.g., mutation and selection) and clarify their role in Algorithm 1. Address missing steps and ensure consistent notations.  Ensure figures and mathematical notations are more accessible.


**3.Clarification on CoEBL vs. Thompson Sampling**

We emphasise that CoEBL is distinct from standard TS. While TS employs stochastic optimism, CoEBL incorporates variation operators for sampling and a selection mechanism for controlling reproduction, effectively embedding evolutionary optimisation principles. This positions CoEBL within the broader Evolutionary Bandit Learning (EBL) category, diverging from conventional bandit algorithms.


**4.Plan for Withdrawal and Resubmission**

Given the detailed and constructive feedback, we will withdraw the paper for now to address the following improvements:

**Motivation for Nash Regret:** Provide a clearer explanation of why Nash regret is the focus of our analysis, emphasising its relevance to game-theoretic learning and practical applications. We will highlight the benefits and limitations of this choice while addressing concerns about convergence guarantees.

**Comparative Analysis:**  Include comparisons with TS and OTS frameworks and address gaps in experimental benchmarks.

**Literature Context:** Add discussions on related works, especially Li et al., and clarify distinctions between CoEBL and TS-based methods.

**Technical and Presentation Enhancements:** Improve the clarity of technical descriptions, figures, and experimental methodologies, including better explanations of evolutionary operators and their role.

We thank the reviewers again for their valuable insights, which will significantly enhance the quality of our work. We look forward to resubmitting a stronger and more comprehensive manuscript soon.

**Withdrawal Confirmation:**

I have read and agree with the venue's withdrawal policy on behalf of myself and my co-authors.